# Young infants display heterogeneous serological responses and extensive but reversible transcriptional changes following initial immunizations

Nima Nouri [1,2,11], Raquel Giacomelli Cao[3,4,11], Eleonora Bunsow[3,11], Djamel Nehar-Belaid [1,11], Radu Marches[1], Zhaohui Xu [3,5], Bennett Smith[3], Santtu Heinonen [3,6], Sara Mertz[3], Amy Leber[7], Gaby Smits[8], Fiona van der Klis[8], Asunción Mejías [3,4,5], Jacques Banchereau[1,9,12], Virginia Pascual [10,12] ✉ & Octavio Ramilo [3,4,5,12] ✉

Infants necessitate vaccinations to prevent life-threatening infections. Our understanding of the infant immune responses to routine vaccines remains limited. We analyzed two cohorts of 2-month-old infants before vaccination, one week, and one-month post-vaccination. We report remarkable heterogeneity but limited antibody responses to the different antigens. Whole-blood transcriptome analysis in an initial cohort showed marked overexpression of interferon-stimulated genes (ISGs) and to a lesser extent of inflammation-genes at day 7, which normalized one month post-vaccination. Single-cell RNA sequencing in peripheral blood mononuclear cells from a second cohort identified at baseline a predominantly naive immune landscape including ISG[hi] cells. On day 7, increased expression of interferon-, inflammation-, and cytotoxicity-related genes were observed in most immune cells, that reverted one month post-vaccination, when a CD8+ ISG[hi] and cytotoxic cluster and B cells expanded. Antibody responses were associated with baseline frequencies of plasma cells, B-cells, and monocytes, and induction of ISGs at day 7.

Infectious diseases represent a major cause of global morbidity and mortality in young children, especially in infants less than 6 months of age[1-4] A growing number of studies have provided insight into early life immune system development[3-10]. Such studies not only uncovered important principles of immune ontogeny, but also highlighted the need for improved, more effective vaccines and anti-infective strategies for this age population. Early life vaccination has become a key determinant for protection against potentially

[1]The Jackson Laboratory for Genomic Medicine, Farmington, CT, USA. [2]Precision Medicine and Computational Biology, Sanofi, 350 Water Street, Cambridge, MA 02141, USA. [3]Center for Vaccines and Immunity, Abigail Wexner Research Institute at Nationwide Children's Hospital, Columbus, OH, USA. [4]Department of Pediatrics, Division of Pediatric Infectious Diseases, Nationwide Children's Hospital, and The Ohio State University College of Medicine, Columbus, OH, USA. [5]Department of Infectious Diseases, St. Jude Children's Research Hospital, Memphis, TN, USA. [6] Pediatric Research Center, New Children's Hospital, Helsinki University Hospital and University of Helsinki, Helsinki, Finland. [7]Department of Laboratory Medicine, Nationwide Children's Hospital, Columbus, OH, USA. [8]National Institute for Public Health and the Environment (RIVM), Bilthoven, The Netherlands. [9]Immunai, New York, NY, USA. [10]Drukier Institute for Children's Health and Department of Pediatrics, Weill Cornell Medicine, New York, NY, USA. [11]These authors contributed equally: Nima Nouri, Raquel Giacomelli Cao, Eleonora Bunsow, Djamel Nehar-Belaid. [12]These authors jointly supervised this work: Jacques Banchereau, Virginia Pascual, Octavio Ramilo. ✉e-mail: vip2021@med.cornell.edu; octavio.ramilo@stjude.org

life-threatening diseases. Following the Centers for Disease Control and Prevention's (CDC) guidelines, young infants are first immunized in the United States against Hepatitis B (HepB), with a vaccine dose given immediately after birth. At 2 months of age (our study population), infants receive the second dose HepB and initial doses of diphtheria, tetanus, acellular pertussis (DTaP), haemophilus influenzae type b (Hib), inactivated poliovirus (IPV), pneumococcal conjugate (PCV13), and rotavirus (RV) vaccines[11,12]. Of these, only the RV vaccine contains a live virus and is administered via oral route, while the remaining vaccines are administered by intramuscular injection and are inactivated.

Qualitative and quantitative differences in immune composition and responses between infants and adults have been reported[3,13–15]. One key difference is the paucity of memory cells in early life[16–18]. Therefore, infants are heavily dependent on both passive transfer of maternal antibodies as well as their own innate immunity to compensate for their immune liability after birth[19]. Meanwhile, the adaptive immune system continues training and evolving as children age. Global immunization programs resulted in a considerable reduction of infant mortality from infectious diseases, but we still have an incomplete understanding of how infants respond to current routine vaccinations. To date, young children require 3 or 4 doses of most vaccines before adequate protection can be achieved. Thus, an in-depth analysis of host immune responses at 2 months of age, which represents priming for most vaccine antigens except HepB, may provide fundamental information for optimization of current and future pediatric vaccination strategies[20].

Analysis of the host transcriptional response has become a valuable tool for dissecting the various pathways involved in the immune response to vaccination in adult populations and, more recently, in children[21–25]. However, only a limited number of transcriptome studies have focused on the initial immune responses in naïve infants. Single-cell RNA sequencing (scRNA-seq) provides opportunities to peer into the immune system at a single-cell resolution and to reveal immune response dynamics at a global scale with concurrent high-level granularity[26–31].

The goal of this study was to take a global look at the longitudinal (time-series study), systems-level analysis of blood immune cells at baseline and upon exposure to the 2-month routine vaccination series. Towards this end, we analyze two independent infant cohorts using whole blood (bulk) transcriptome analysis for the first cohort, and single peripheral blood mononuclear cells (PBMC) RNA-seq for the second cohort. In combination with serological assays, our analysis offers a comprehensive insight into the coordinated changes that occur in a predominantly naïve infant immune system upon its initial exposure to a diverse range of microbial antigens.

## Results

### Routine vaccinations in 2-month-old infants elicit heterogenous antibody responses

We first analyzed antibody titers in a cohort of otherwise healthy full-term 2-month-old infants (Cohort 1 Bulk Transcriptome; $n = 19$) (Supplementary Data 1) during their routine 2-month vaccinations. Samples were obtained before vaccination (day 0) and approximately 1 week (day 7) and 1 month later (day 30) (demographic characteristics can be found in Supplementary Data 1). At baseline, the majority of infants had low antibody titers against most pneumococcus serotypes, Hemophilus influenzae type b (Hib), and diphtheria (Supplementary Fig. S1a and Supplementary Data 1). Antibodies against tetanus and pertussis antigens were however detectable, most likely reflecting vaccination with Tdap during pregnancy. The 4-week post-vaccination antibody responses were heterogeneous. Thus, the number of infants and rates of seroprotection for PCV13 (defined by the response to at least four antigens), Hib, and DTaP at day 30 were 7 (38%), 2 (13%), and 11 (60%), respectively.

Next, we analyzed an independent cohort (Cohort 2 scRNA-seq; $n = 6$). Following a similar study design, we collected longitudinal blood samples at three time points (total $n = 18$ samples) in the context of the 2-month routine vaccinations (demographic characteristics can be found in Supplementary Data 1). Antibody responses were determined in 5/6 infants (I1, I2, I4, I5, and I6) with available paired serum samples before vaccination (time point A) and 4 weeks later (time point C; Supplementary Fig. S1b and Supplementary Data 1). I3 samples were not available due to low blood draw volume. The antibody responses were measured against slightly different vaccine antigens compared to the initial cohort (Supplementary Fig. S1b). All infants had received the first dose of Hepatitis B vaccine (recombinant protein) at birth and showed low titers (<0.8 Miu/mL) at time point A. After the second dose (booster), all five infants responded with 8- to 70-fold increase in titers at time point C. Responses to the remaining antigens were heterogeneous. Only one infant (I6) showed a weak serologic response (2-fold increase in titer) to Diphtheria (toxoid) vaccination at time point C. For tetanus (toxoid), all 5 infants had titers higher than 0.1 IU/mL prior to immunization (likely maternal antibodies transferred across the placenta) that failed to increase post-immunization. The pneumococcus conjugate polysaccharide vaccine (PCV13) induced variable responses to the 13 different antigens. No serological response was observed in infant I4. I1 responded to only 1 antigen, I2 to 2 antigens and I5 to 3 antigens. Finally, I6 responded to 10 pneumococcal antigens. The orally administered live rotavirus vaccine induced robust serological IgA responses in 4/5 infants at time point C, with I5 being the exception. In summary, 2-month-old infants showed good responses to the second dose of Hepatitis B and highly heterogeneous responses to the remaining vaccines. They were ranked accordingly into three categories. (1) Best-Responder (Best-R): I6 fully responded to Hepatitis B, Diphtheria, Pneumococcus, and Rotavirus. (2) Good-Responders (Good-Rs): I1 and I2 fully responded to Hepatitis B and Rotavirus, and partially responded to Pneumococcus. (3) Weak-Responders (Weak-Rs): I4 fully responded to Hepatitis B and Rotavirus; I5 fully responded to Hepatitis B and partially to Pneumococcus.

### Post-vaccine whole blood transcriptome analysis demonstrates robust innate immune responses

We analyzed whole blood gene expression profiles in the initial 2-month-old infant cohort. Samples available for analysis were obtained at day 0 ($n = 24$; baseline), day 7 ($n = 23$), and day 30 ($n = 21$). Compared to the baseline, we identified 419 differentially expressed transcripts (Supplementary Data 2), 76% of which were overexpressed on day 7 after vaccination (adjusted $p$ value < 0.05) (Fig. 1a). To elucidate the biological pathways elicited by the vaccines, we applied an established modular analysis tool[32,33]. Transcripts overexpressed at day 7 included interferon-stimulated genes (ISGs) (modules M1.2, M3.4, and M5.12), as well as monocytes (M4.14), inflammation (M3.2, M4.2, M4.6, M5.1), and plasma cells (PC) (M4.11) related genes (Fig. 1b). Conversely, we observed underexpression of T-cell and B-cell-related modules (M4.1/M4.15 and M4.10 respectively). On day 30, we observed overexpression of B cell- and PC-related modules (M4.10 and M4.11, respectively) (Fig. 1b), while expression of ISGs and most inflammation-related genes had returned to baseline. While no significant variations in the transcriptome of neutrophil genes were observed at day 7, this shifted to underexpression on day 30, with 13% of genes showing reduced expression (Fig. 1b).

### scRNA-seq of PBMCs from vaccinated infants identifies 19 distinct cell clusters

scRNA-seq was performed using a droplet-based single-cell 10× Chromium platform[34]. Samples were obtained at baseline prior to vaccine administration (time point A), 8.6 ± 2.4 (mean ± SD) days post-vaccination (time point B), and 32.2 ± 2.9 (mean ± SD) days post-vaccination (time point C) (Fig. 2a, b and Supplementary Fig. S2a).

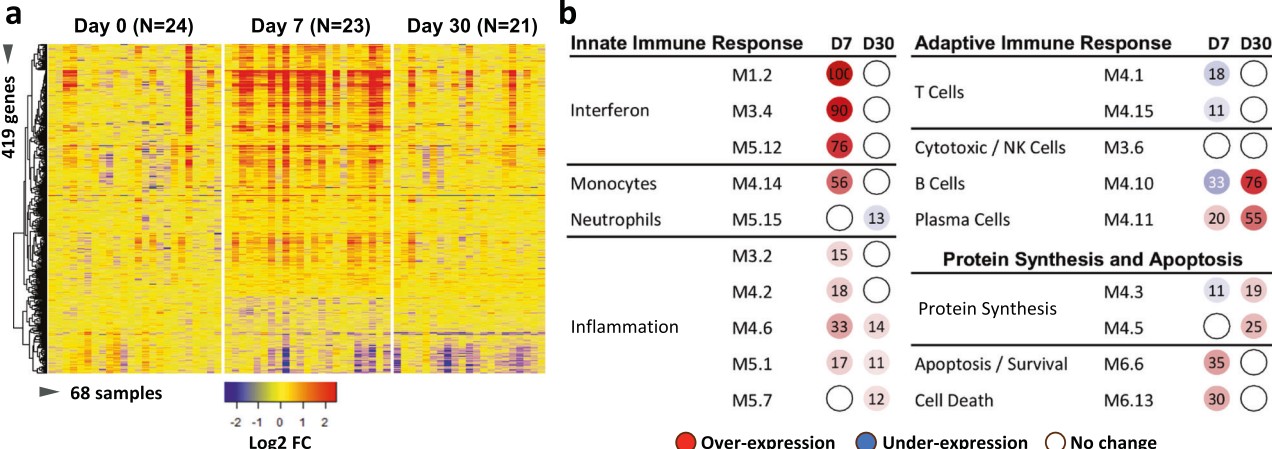

**Fig. 1 | Elevated expression of interferon, inflammatory, and plasma cell-associated modules revealed by bulk transcriptome analysis. a** Heatmap of bulk transcriptome expression comparing day 0 ($n = 24$ infants), day 7 ($n = 23$ infants), and day 30 ($n = 21$ infants). $N = 419$ differentially expressed genes at day 7 are shown (rows) across the three time points (columns). Differentially expressed genes were identified using the Kruskal–Wallis H test with an adjusted $p$ value < 0.05. The color legend indicates the log2-fold change. **b** Expression patterns of gene modules. Day 0 (baseline) serves as the reference, and changes in gene module expression are depicted using a color scale: red indicating overexpression, blue indicating underexpression, and white indicating no significant difference compared to day 0. The value in each circle indicates the percentage of significant module-related genes showing underexpression (blue) or overexpression (red).

After filtration and quality control, the raw data were merged, normalized, and batch corrected (Supplementary Fig. S2b). This analysis yielded 69,802 PBMCs, with $6400 \pm 2428$ (mean $\pm$ SD) reads/cell and $1725 \pm 464$ (mean $\pm$ SD) genes/cell (see Methods and Supplementary Fig. S2c, d). Unsupervised clustering followed by marker identification yielded 19 cell clusters, which could be assigned to 10 cell types including single clusters of megakaryocytes (Mgks), plasma cells (PCs), classical dendritic cells (cDCs), plasmacytoid DCs (pDCs), and erythroid cells (Eryts); two clusters of natural killer cells (NKCs), monocytes (Mono: CD14+ and CD16+) and B cells (BCs); three clusters of CD8+ T cells, and five clusters of CD4+ T cells (Fig. 2c, d). Automated reference-based[35] annotation using Azimuth web application[36] supported the overall outcomes of the manual annotation (Supplementary Fig. S3a). Overall, cell numbers varied from 54 Mgks to 47,064 CD4+ T cells (Fig. 2e, f) with an unbiased distribution of cells across batches (10× runs), samples, and infants (Supplementary Figs. S2e and S3b).

**Vaccination induces a universal and transient interferon response in CD4+ T cells**

CD4+ T cells represented the major immune compartment (67% of PBMCs; 47,064 cells) (Fig. 3a). Five subclusters (SCs) were identified: CD4_T$_{naive}$ (22,611 cells); CD4_T$_{AREG+}$ (17,133 cells); CD4_T$_{ISGhi}$ (2946 cells); CD4_T$_{MAIT}$ (2265 cells); and CD4_T$_{REG}$ (2109 cells) (Supplementary Fig. S4a). The majority of CD4+ T cells consisted of naive cells (Fig. 3b). Thus, SCs CD4_T$_{naive}$, CD4_T$_{AREG+}$, and CD4_T$_{ISGhi}$ expressed naïve/central memory markers including *CCR7*, *LEF1* and *SELL*, while CD4_T$_{MAIT}$ SC exhibited *S100A4* suggesting a memory phenotype. CD4_T$_{REG}$ SC included cells expressing naïve and memory markers. CD4_T$_{naive}$ SC expressed *SOX4*, the neonatal IgG Fc receptor *FCGRT* and *GABPB1-AS1*, a LncRNA that downregulates GABPB1, a master regulator of nuclear-encoded mitochondrial genes. CD4_T$_{AREG+}$ SC expressed *AREG* (Amphiregulin); a cytokine involved in tissue repair. CD4_T$_{ISGhi}$ SC expressed the highest levels of ISGs (ISGhi), including *ISG15* (negative regulator of type I IFN signaling), *IFI6*, and *IFI44L* (Fig. 3f) and was the main CD4+ T cell SC expanded at time point B (Fig. 3d) in all six infants (Fig. 3e and Supplementary Fig. S4b). CD4_T$_{naive}$, CD4_T$_{AREG+}$, and CD4_T$_{ISGhi}$ SCs expressed transcripts encoding cytotoxic molecules such as *GZMA* and *GZMM*. CD4_T$_{MAIT}$ SC exhibited MAIT (Mucosa-associated invariant T) and/or Th17-related markers (e.g., *KLRB1*, *ZBTB16* and *CCR6*), cytotoxic markers (*GZMA* and

*GZMK*), several members of the Annexin family, *DDIT4* (*REDD-1*), an inhibitor of mTORC1 that plays a role in T cell proliferation and survival[37,38], as well as *GPR183* (Fig. 3c), a chemoattractant receptor that directs CD4+ T cells to the paracortical region of lymph nodes to enable antigen recognition, differentiation, effector activity, and CD8+ T cell help[39]. None of the SCs had a detectable expression of the canonical transcription factors (TF) for TH1 (*TBX21*) or TH17 (*RORC*) cells, but the TH2 TFs *GATA*-3 and signal transducer and activator of transcription (*STAT*) 6 were present in all SCs, and were especially upregulated in CD4_T$_{MAIT}$ SC. CD4_T$_{REG}$ SC included regulatory T cell-related transcripts, such as *TIGIT* and *IL2RA*, as well as the TFs *FOXP3* and IKZF2[40]. The chemokine receptor CXCR5, a hallmark of follicular T helper cells (Tfh), was not detected in any SC (Fig. 3b).

As vaccination induces an antiviral-like response involving both the myeloid and lymphoid compartments, we calculated interferon (IFN) scores (average expression of 137 ISGs derived from previously defined IFN transcriptional modules, which correspond to sets of genes coordinately expressed in different diseases and defined as specific groups[32,33] in each individual cell. The IFN scores showed a significant ($p < 0.05$) increase at time point B and a return to baseline at point C across all CD4 SCs, indicating their transient induction postvaccination (Fig. 3g). Of note, CD4_T$_{ISGhi}$ SC showed increased baseline ISG expression levels, suggesting tonic activation of the IFN pathway in specific cell subsets in these infants, as previously reported in older children[41].

Differential expression (DE) analysis (using time point A/pre-vaccination as baseline) identified 1147 significantly modulated genes (sDEGs: $p < 0.05$, log2-fold-change > 0.25) over all CD4 SCs (Fig. 3h and Supplementary Data 3). As expected, the sDEGs were predominantly identified at time point B (1133/1147), whereas only a few remained at time point C (35/1147) (Fig. 3h). Most of the time point B sDEGs (up-/downregulated) were identified in SC CD4_T$_{AREG+}$ (Supplementary Fig. S4c). Enrichment analysis (see Methods) of up- and downregulated sDEGs across the CD4+ T SCs at time points B and C confirmed that IFN response modules were positively correlated with upregulated sDEGs at time point B across CD4 SCs (Supplementary Fig. S4d). In fact, 73/1147 sDEGs were ISGs, and the expression of 71 of them increased at time point B in at least one of the SCs (Fig. 3i).

Overall, this analysis reveals that, as expected, the baseline blood CD4+ T cell compartment of 2-month-old infants is predominantly naive. Surprisingly, however, most naïve SCs express

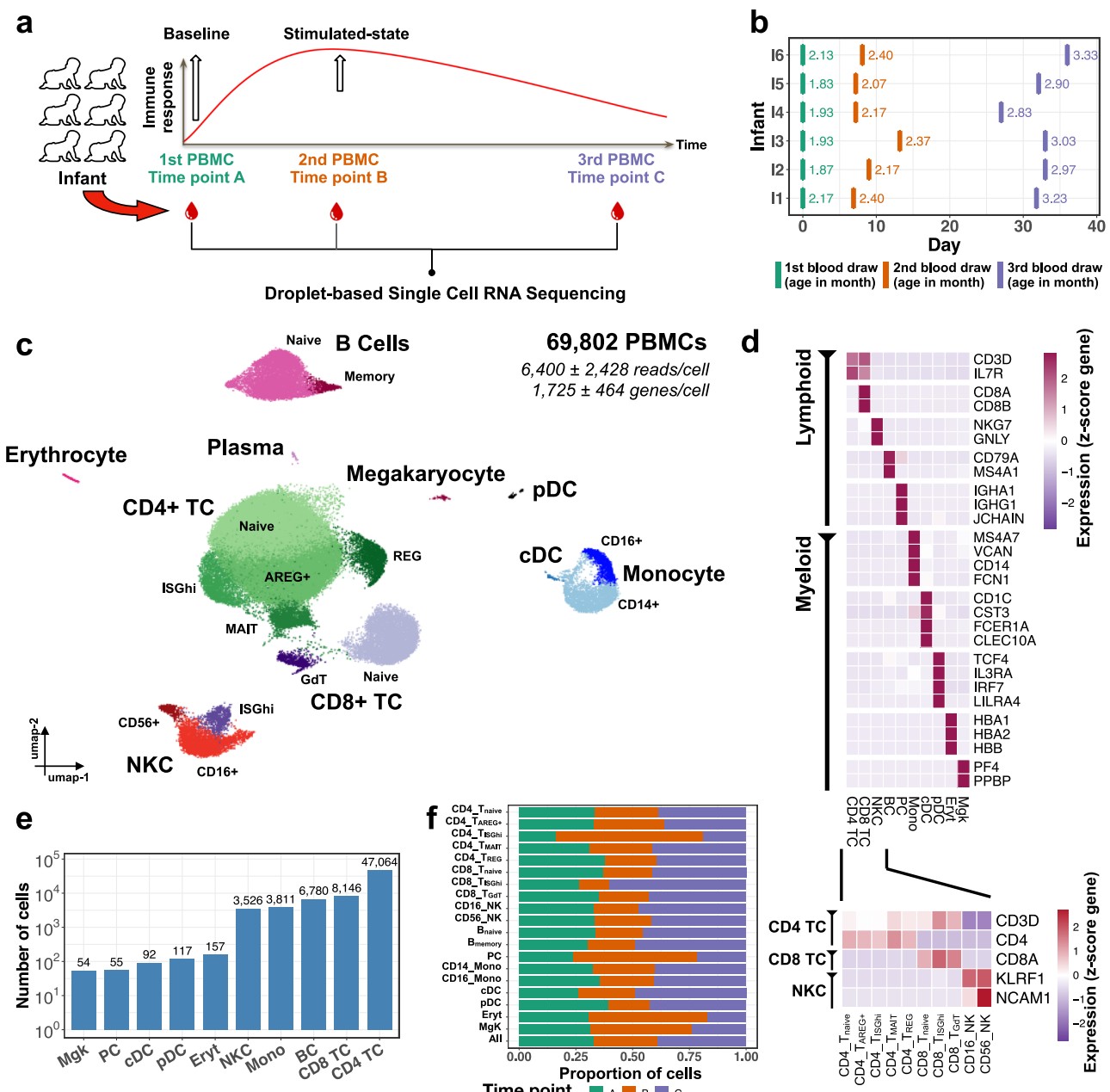

**Fig. 2 | Longitudinal single-cell profiling of vaccinated infants. a** Overview of the study design illustrating the longitudinal sampling of peripheral blood mononuclear cells (PBMC) from a cohort of vaccinated infants ($n = 6$). **b** Sampling time point in days (x-axis) for each infant (y-axis), with labels indicating infants' ages in months at the time of sampling. **c** UMAP representation of 69,802 PBMCs (obtained from $n = 6$ vaccinated infants and grouped into 19 clusters) by scRNA-seq, color-coded for the indicated cell types (CD4 TC: CD4+ T cells, CD8 TC: CD8+ T cells, NKC: natural killer cells, BC: B cells, PC: plasma cells, Mono: monocytes, cDC: conventional dendritic cells, pDC: plasmacytoid dendritic cells, Eryt: erythroid cells, Mgk: megakaryocyte). **d** Feature plots showing marker genes for each of the different cell types. **e** Number of cells per cell type. **f** Proportion of cells per time point for each of the different cell type subclusters.

cytotoxic molecules. While T regulatory markers were also detected, neither TH1, TH17 nor Tfh markers could be identified either at baseline or post-vaccination. Importantly, the infant's CD4+ T cells mounted a transient and universal ISG response 1 week post-vaccination that returned to baseline 1 month later. Furthermore, our study identified a blood CD4+ T cell subset with increased baseline ISG expression at this early age.

### Post-vaccination CD8+ T cells demonstrate interferon and cytotoxic transcriptional changes
scRNA-seq analysis yielded 8146 CD8+ T cells, the second largest immune compartment (12% of PBMCs) (Fig. 4a), which was composed

of three subclusters (SCs): CD8_T$_{naive}$ (6609 cells); CD8_T$_{ISGhi}$ (895 cells); and CD8_T$_{GdT}$ (642 cells) (Supplementary Fig. S5a). CD8_T$_{naive}$ and CD8_T$_{GdT}$ expressed naïve/central memory T-cell markers including *CCR7*, *LEF1* and *SELL*, while CD8_T$_{ISGhi}$ SC exhibited *S100A4*, suggesting a memory phenotype (Fig. 4b). CD8_T$_{ISGhi}$ showed the highest expression level of ISGs (ISG$^{hi}$), together with upregulation of cytotoxic transcripts, including *PRF1, KLRG1, GZMA, GZMB*, and *GZMH* (Fig. 4b). In addition, the CD8_T$_{ISGhi}$ SC exhibited T$_{EMRA}$ (terminal effector memory CD8+ T cells) related markers such as *CX3CR1*, *FGFBP2*, and *FCGR3A*[42]. Interestingly, this SC was expanded at time point C (Fig. 4d) in the majority (4/6) of infants (Fig. 4e and Supplementary Fig. S5b), possibly reflecting either a prolonged response to

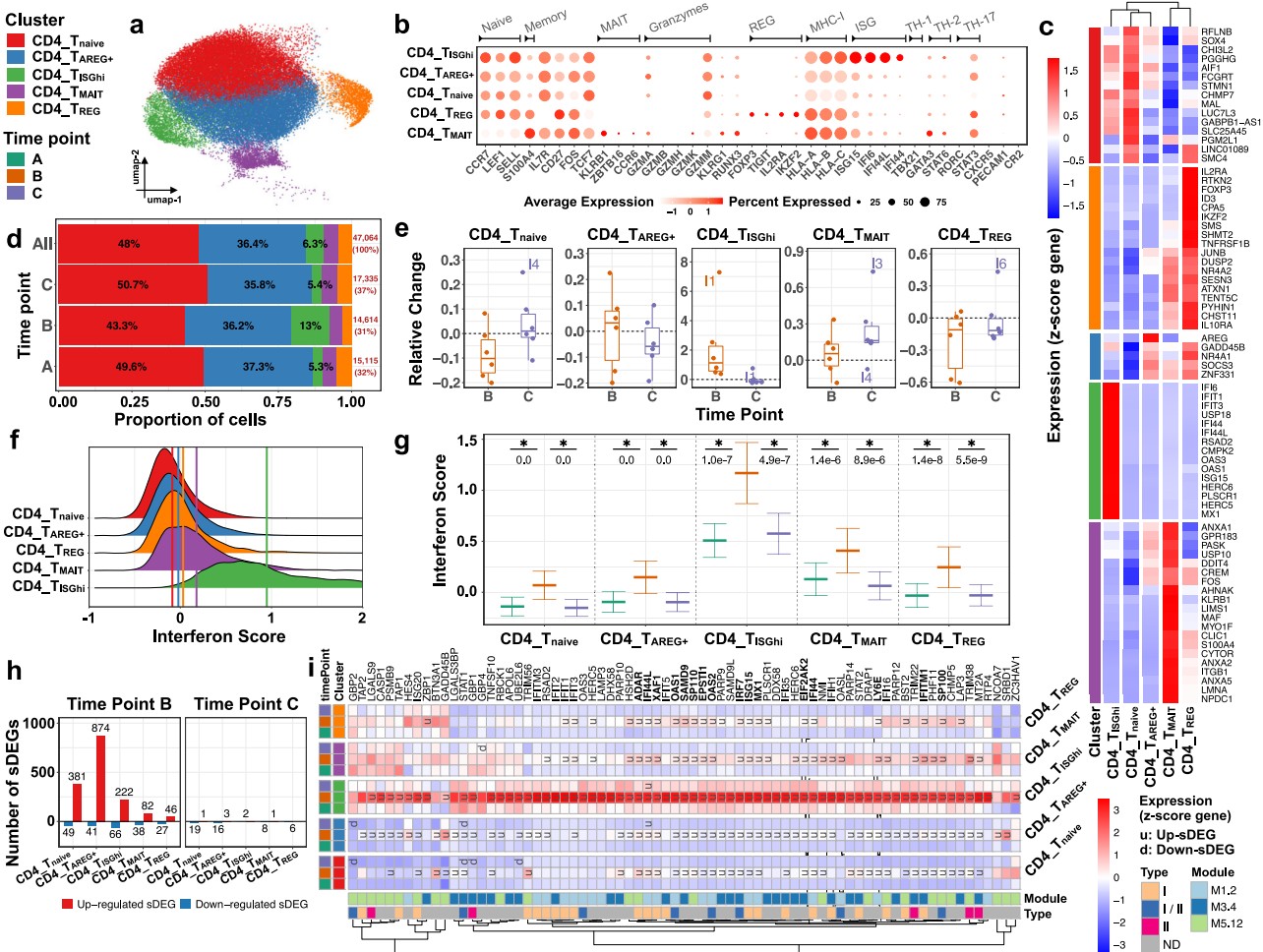

**Fig. 3 | Infants' CD4+ T cells exhibit transient and universal ISG response 1 week post-vaccination.** Subcluster (SC) and time point color codes are shown in the top-left. **a** UMAP plot representing n = 5 color-coded SCs encompassing 47,064 cells. **b** Expression levels of selected genes for each SC. **c** Heatmap representing scaled expression values of the top 15 differentially expressed genes defining each of the SCs. The differentially expressed genes are determined using two-sided Wilcoxon rank-sum tests, with adjusted p values less than 0.05. **d** Proportion of cells (labels) per time point for each SC. **e** Relative change of cell abundance at post-vaccination (time points B and C) compared to the baseline (time point A) for each SC. The results are depicted in boxplots, in which the value for each infant (total n = 6) is represented by a dot; the upper and lower bounds represent the 75% and 25% percentiles, respectively. The center bars indicate the medians, and the whiskers denote values up to 1.5 interquartile ranges above 75% or below the 25% percentiles. Data beyond the end of the whiskers are outliers (labeled infants). **f** Interferon score

distribution for each SC ranked from the lowest (top) to highest (bottom). Color-coded vertical lines indicate the average score per SC. **g** Interferon score calculated for each cell per time point and SC. Error bar indicates standard deviation calculated across cells. Horizontal bar indicates mean. Asterisk (*) indicates a significant difference between two consecutive time points, determined using two-sided t-tests, with unadjusted p < 0.05 (labels). **h** Number of significantly differentially expressed genes (sDEGs) at post-vaccination (time points B and C) and regulation-type (up-/downregulated) compared to the baseline (time point A) for each SC. **i** Heatmap representing scaled expression values of significantly differentially expressed interferon genes per time point for each SC. Genes upregulated at time point B across all SCs are shown in bold. The color keys on the left and bottom represent gene associations. sDEGs are determined as those with a log2-fold-change > 0.25 and adjusted p < 0.05, calculated using two-sided Wilcoxon rank-sum tests.

the live Rotavirus vaccine, or to multiple vaccines. Finally, CD8_T$_{GdT}$ SC, and to a lesser extent CD8_T$_{ISGhi}$ SC, included GdT (Gamma delta) related markers such as *TRDC*, *TRGC2* and *TRG-AS1* (Fig. 4c).

IFN and cytotoxic scores, calculated based on the average expression of ISGs (n = 137) and cytotoxic-RGs (n = 105) from previously described modules in each individual cell (see Methods), were highest amongst cells from CD8_T$_{ISGhi}$ SC (Fig. 4f) both at baseline and post-vaccination (Fig. 4f, g). A significant (p < 0.05) increase in IFN scores at time point B was observed across all CD8+ T cell SCs (Fig. 4g, left panel) and returned to baseline at time point C. Cytotoxic scores in CD8_T$_{naive}$ and CD8_T$_{GdT}$ SCs remained unchanged over time. CD8_T$_{ISGhi}$ SC expressed the highest levels of cytotoxic transcripts over all three time points (cytotoxic$^{hi}$), decreasing significantly (p < 0.05) at time point B (Fig. 4f, g, right panel) and returning to baseline at time point C. Intriguingly, when we compared the IFN

scores of total CD4+ and CD8+ T cells, CD4+ T cells expressed lower levels (p < 0.05) of ISGs at time points A and C, but higher levels at time point B (Supplementary Fig. S5c).

Vaccine-induced signatures mostly faded around 1 month post-vaccination. At time points B and C, differential expression analysis identified significant changes in 358 genes (sDEGs: p < 0.05, log2-fold-change > 0.25) over all CD8 SCs (Fig. 4h and Supplementary Data 3). Of the 358 sDEGs, 319 were identified at time point B, and only 54 at time point C (Fig. 4h). Most of the time point B, sDEGs mapped to the CD8_T$_{naive}$ SC (Supplementary Fig. S5d). Enrichment analysis (see Methods) showed that most IFN response modules were positively correlated with upregulated sDEGs at this time point across all CD8 SCs (Supplementary Fig. S5e), with 44/358 sDEGs being ISGs (Fig. 4i). At time point C, the overexpressed genes in CD8_T$_{ISGhi}$ SC were mostly cytotoxic and MHC class II transcripts.

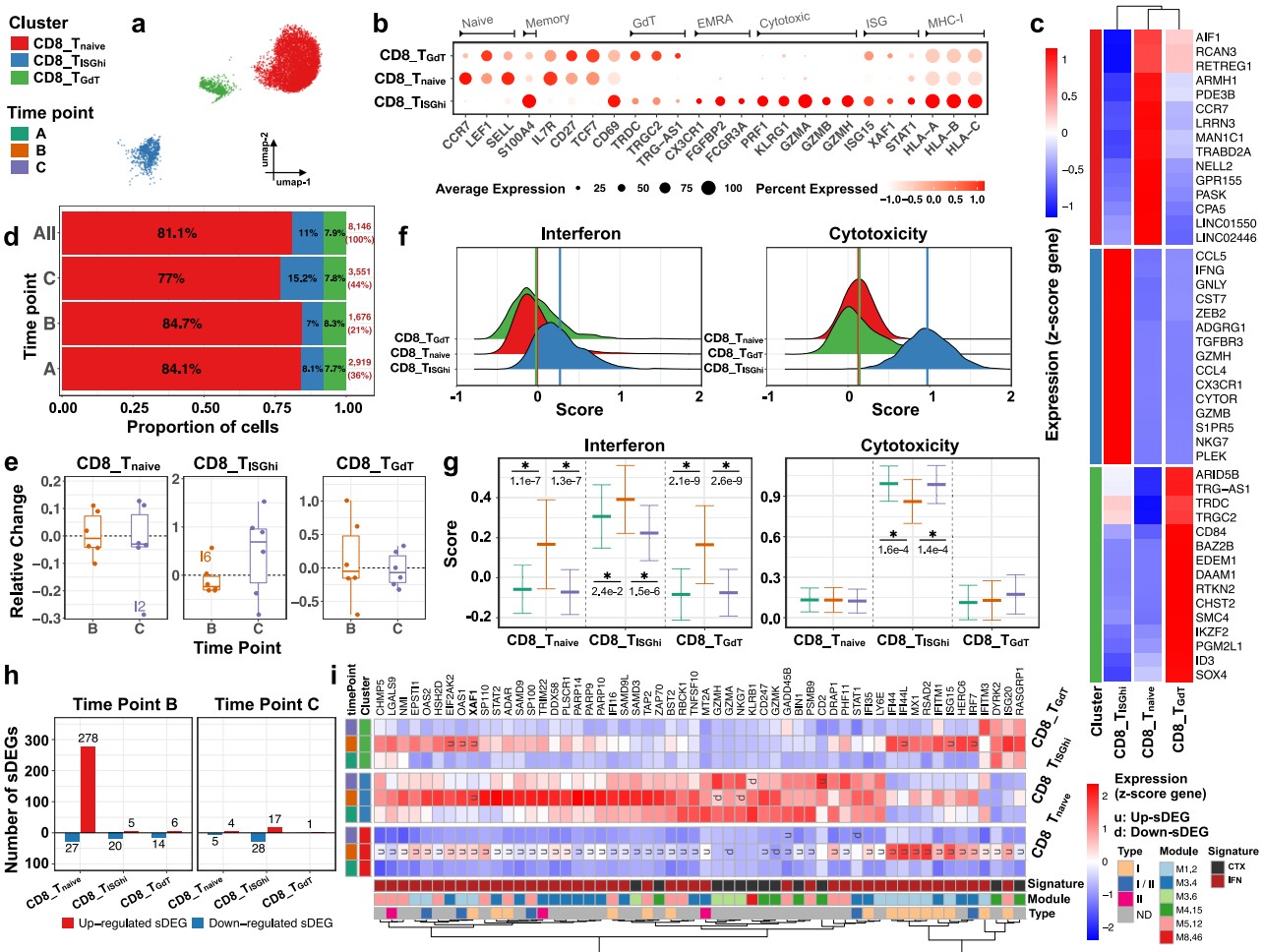

**Fig. 4 | Infants' T_EMRA-like CD8+ T cells expand post-vaccination, revealing heightened interferon and cytotoxic activity.** Subcluster (SC) and time point color codes are shown in the top-left. **a** UMAP plot representing $n = 3$ color-coded SCs encompassing 8146 cells. **b** Expression levels of selected genes for each SC. **c** Heatmap representing scaled expression values of the top 15 differentially expressed genes defining each of the SCs. The differentially expressed genes are determined using two-sided Wilcoxon rank-sum tests, with adjusted $p$ values less than 0.05. **d** Proportion of cells (labels) per time point for each SC. **e** Relative change of cell abundance at post-vaccination (time points B and C) compared to the baseline (time point A) for each SC. The results are depicted in boxplots, in which the value for each infant (total $n = 6$) is represented by a dot; the upper and lower bounds represent the 75% and 25% percentiles, respectively. The center bars indicate the medians, and the whiskers denote values up to 1.5 interquartile ranges above 75% or below the 25% percentiles. Data beyond the end of the whiskers are outliers (labeled infants). **f** Interferon score distribution (left panel) and cytotoxicity score distribution (right panel) for each SC ranked from the lowest (top) to highest (bottom). Color-coded vertical lines indicate the average score per SC. **g** Interferon score (left panel) and cytotoxicity score (right panel) calculated for each cell per time point and SC. Error bar indicates standard deviation calculated across cells. Horizontal bar indicates mean. Asterisk (*) indicates a significant difference between two consecutive time points, determined using two-sided $t$-tests, with unadjusted $p < 0.05$ (labels). **h** Number of significantly differentially expressed genes (sDEGs) at post-vaccination (time points B and C) and regulation-type (up-/downregulated) compared to the baseline (time point A) for each SC. **i** Heatmap representing scaled expression values of significantly differentially expressed interferon and cytotoxicity-related genes per time point for each SC. Genes upregulated at time point B across all SCs are shown in bold. The color keys on the left and bottom represent gene associations. sDEGs are determined as those with a log2-fold-change > 0.25 and adjusted $p < 0.05$, calculated using two-sided Wilcoxon rank-sum tests.

Thus, the circulating CD8+ T cells of 2-month-old infants comprise distinct subpopulations, including T_EMRA-like cells. Administration of multiple vaccines resulted in significant transcriptional changes in CD8+ T cells that returned to baseline at 1 month, and in the expansion at this time of an ISG^hi, cytotoxic^hi T_EMRA-like population.

**Vaccination induces expansion of a CXCR5- naive B cell subcluster and increased expression of interferon-stimulated genes**

scRNA-seq analysis yielded 6780 B cells, the third largest compartment (10% of PBMCs) (Fig. 5a) comprising five subclusters (SCs): B_naive (3012 cells), B_CCR7- (1862 cells), B_CXCR5- (1069 cells), B_memory (712 cells), and B_ISGhi (125 cells) (Supplementary Fig. S6a). B cells mainly consisted of naive cells, including B_naive, B_CCR7-, B_CXCR5-, and B_ISGhi SCs, which expressed *IGHD* (IgD) and *IGHM* (IgM) and lacked *CD27* (Fig. 5b).

B_memory expressed the memory marker CD27 together with switched isotypes *IGHG1* and *IGHA1*. B_ISGhi SC (the smallest BC SC) showed the highest expression of ISGs (Fig. 5c, f) and was expanded at time point B in 2/6 infants (Fig. 5d, e and Supplementary Fig. S6b). B_ISGhi SC and B_naive SC expressed *CCR7* and *CXCR4*, which guide B cells to T cell areas in lymphoid organs and bone marrow, respectively. All SCs except for B_CXCR5- expressed CXCR5, which provides signals for trafficking to lymphoid follicles. In addition to lacking expression of CXCR5, B_CXCR5- SC expressed *FCRL2* and *5*, which are markers of extra-follicular B cells[43]. This SC also expressed *CD180*, a pathogen receptor involved, together with TLR4, in recognition of lipopolysaccharide (LPS) from Gram-negative bacteria[44]. Interestingly, B_CXCR5- SC was expanded at time point B in 3/6 infants. At time point C, and compared with time point B, we also observed increased frequencies of B_naive and B_memory cells (Fig. 5e).

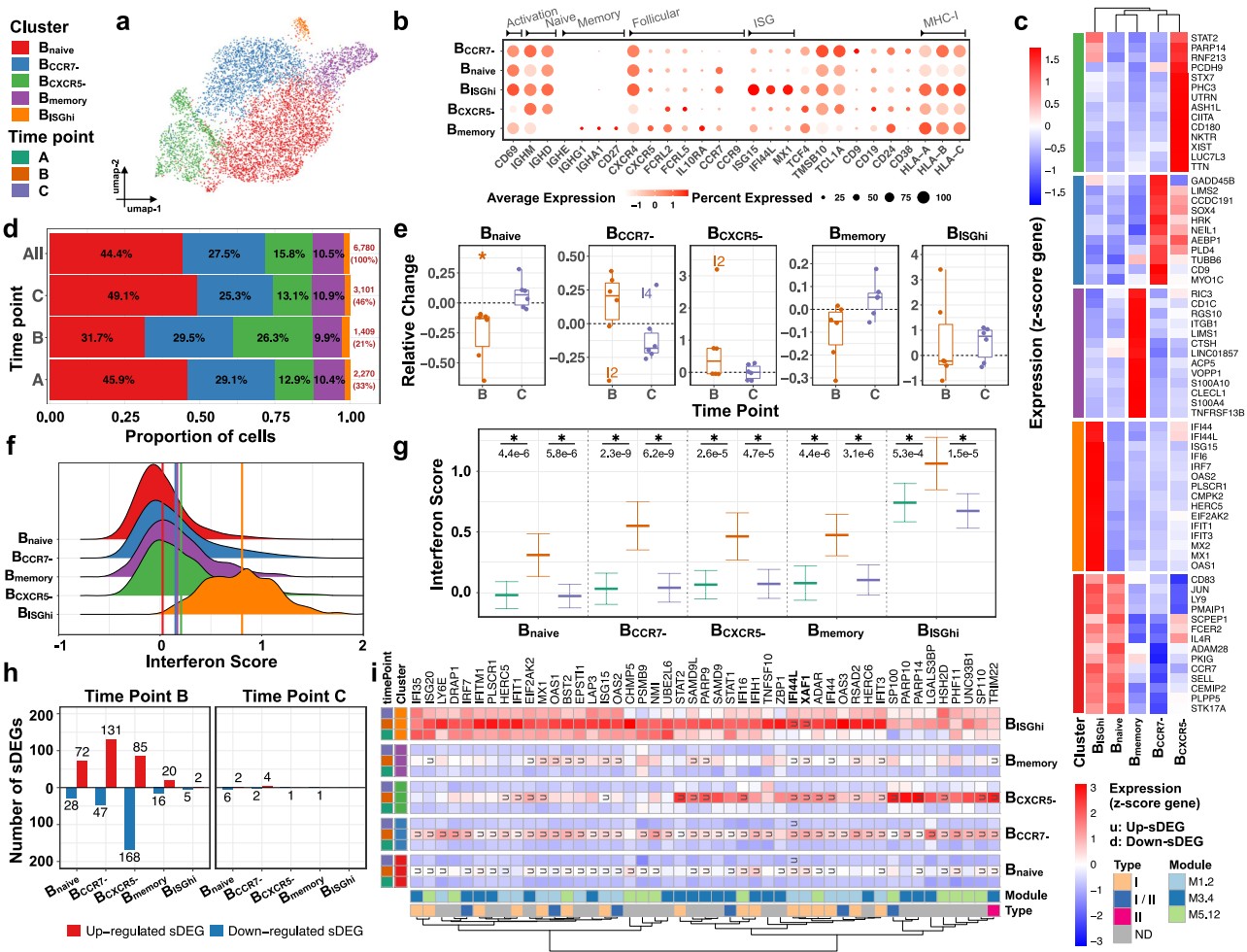

**Fig. 5 | Infants' B cells exhibit transient and universal ISG response 1 week post-vaccination.** Subcluster (SC) and time point color codes are shown in the top-left. **a** UMAP plot representing $n = 5$ color-coded SCs encompassing 6780 cells. **b** Expression levels of selected genes for each SC. **c** Heatmap representing scaled expression values of the top 15 differentially expressed genes defining each of the SCs. The differentially expressed genes are determined using two-sided Wilcoxon rank-sum tests, with adjusted $p$ values less than 0.05. **d** Proportion of cells (labels) per time point for each SC. **e** Relative change of cell abundance at post-vaccination (time points B and C) compared to the baseline (time point A) for each SC. The results are depicted in boxplots, in which the value for each infant (total $n = 6$) is represented by a dot; the upper and lower bounds represent the 75% and 25% percentiles, respectively. The center bars indicate the medians, and the whiskers denote values up to 1.5 interquartile ranges above 75% or below the 25% percentiles. Data beyond the end of the whiskers are outliers (labeled infants). **f** Interferon score

distribution for each SC ranked from the lowest (top) to highest (bottom). Color-coded vertical lines indicate the average score per SC. **g** Interferon score calculated for each cell per time point and SC. Error bar indicates standard deviation calculated across cells. Horizontal bar indicates mean. Asterisk (*) indicates a significant difference between two consecutive time points, determined using two-sided $t$-tests, with unadjusted $p < 0.05$ (labels). **h** Number of significantly differentially expressed genes (sDEGs) at post-vaccination (time points B and C) and regulation-type (up-/downregulated) compared to the baseline (time point A) for each SC. **i** Heatmap representing scaled expression values of significantly differentially expressed interferon-related genes per time point for each SC. Genes upregulated at time point B across all SCs are shown in bold. The color keys on the left and bottom represent gene associations. sDEGs are determined as those with a log2-fold-change > 0.25 and adjusted $p < 0.05$, calculated using two-sided Wilcoxon rank-sum tests.

Consistent with other cell types, the IFN scores calculated based on the average expression of ISGs ($n = 137$) in each single cell showed a significant ($p < 0.05$) increase at time point B across all BC SCs compared to baseline, indicating induction post-vaccination (Fig. 5g) that returned to baseline at time point C. Of note, $B_{ISGhi}$ SC showed higher ISG expression levels at all three time points, indicating tonic or constitutive activation of the IFN pathway.

Differential expression analysis revealed 383 vaccine-induced transcripts (sDEGs: $p < 0.05$, log2-fold-change > 0.25) (Supplementary Data 3) that were predominantly identified at time point B (375/383) (Fig. 5h), from which 198 genes were identified in $B_{naive}$, $B_{CCR7-}$, and $B_{CXCR5-}$ SCs (Supplementary Fig. S6c). Enrichment analysis (see Methods) indicated that a subset of upregulated sDEGs (47/383) at time point B across all SCs were ISGs (Supplementary Fig. S6d and Fig. 5i).

Thus, 90% of the 2-month-old infant's B cell compartment is composed of unswitched, transitional and/or naive B cells, including both ISG$^{hi}$ and CXCR5- SCs, while 10% represent switched memory B cells. Vaccination induced an expansion at day 7 of a CXCR5- naive SC in 50% of the patients and a broad and substantial increase in ISG expression levels. At 4 weeks after vaccination, we also observed moderate expansion of naïve and memory B cell frequencies.

scRNA-seq analysis yielded 55 Plasmablasts/Plasma cells (0.08% of PBMCs). At baseline, infant's PCs were characterized by the expression of *IGHM* (IgM), *IGHG1* (IgG), and *IGHA1* (IgA), together with *CD27*, *PRDM1* (*Blimp-1*), *XBP1*, and *MZB1* (Supplementary Fig. S7a). At time point B, PC frequencies expanded, and interferon scores increased significantly ($p < 0.05$) (Supplementary Fig. S7b), returning to the baseline at time point C. We observed, however, an increased expression of discrete sets of ISGs at each time point post-vaccination

(Supplementary Fig. S7c). Interestingly, while few infant PCs expressed *CCR9*, a chemokine receptor associated with intestinal migration, up to half of them expressed *CCR10*, a homing receptor for skin and intestine (Supplementary Fig. S7a). Thus, 2-month-old infants display an expansion of PCs expressing ISGs together with skin and intestine homing receptors at day 7 post-vaccination, returned to baseline at time point C.

## Vaccination alters the monocyte transcriptional landscape of infants with marked expression of interferon-stimulated and inflammation-related genes

scRNA-seq analysis yielded 3811 monocytes (Mono), the fourth largest immune compartment (5.5% of PBMCs) (Fig. 6a). Two

subclusters (SCs) were identified: CD14_Mono (2502 cells) and CD16_Mono (1309 cells) (Supplementary Fig. S8a). CD14_Mono was characterized by the expression of CD14 (CD14+ monocytes) and *S100A8/9* (Fig. 6b, c). CD16_Mono expressed FCGR3A (CD16+ monocytes) and the non-classical/intermediate monocyte marker *MS4A7* (Fig. 6b). Pro-inflammatory cytokine genes *CXCL8* (IL8) and IL1B were detected in both SCs, albeit to a higher degree in CD14_Mono. The frequency of CD14_Mono fluctuated among infants, with a minimal expansion observed in five infants at time point B post-vaccination but, on average, monocytes frequency remained stable through the study (Fig. 6d, e and Supplementary Fig. S8b).

IFN and inflammation (INFLM) scores (based on the expression of 137 ISGs and 334 INFLM-regulated genes were calculated. CD14_Mono

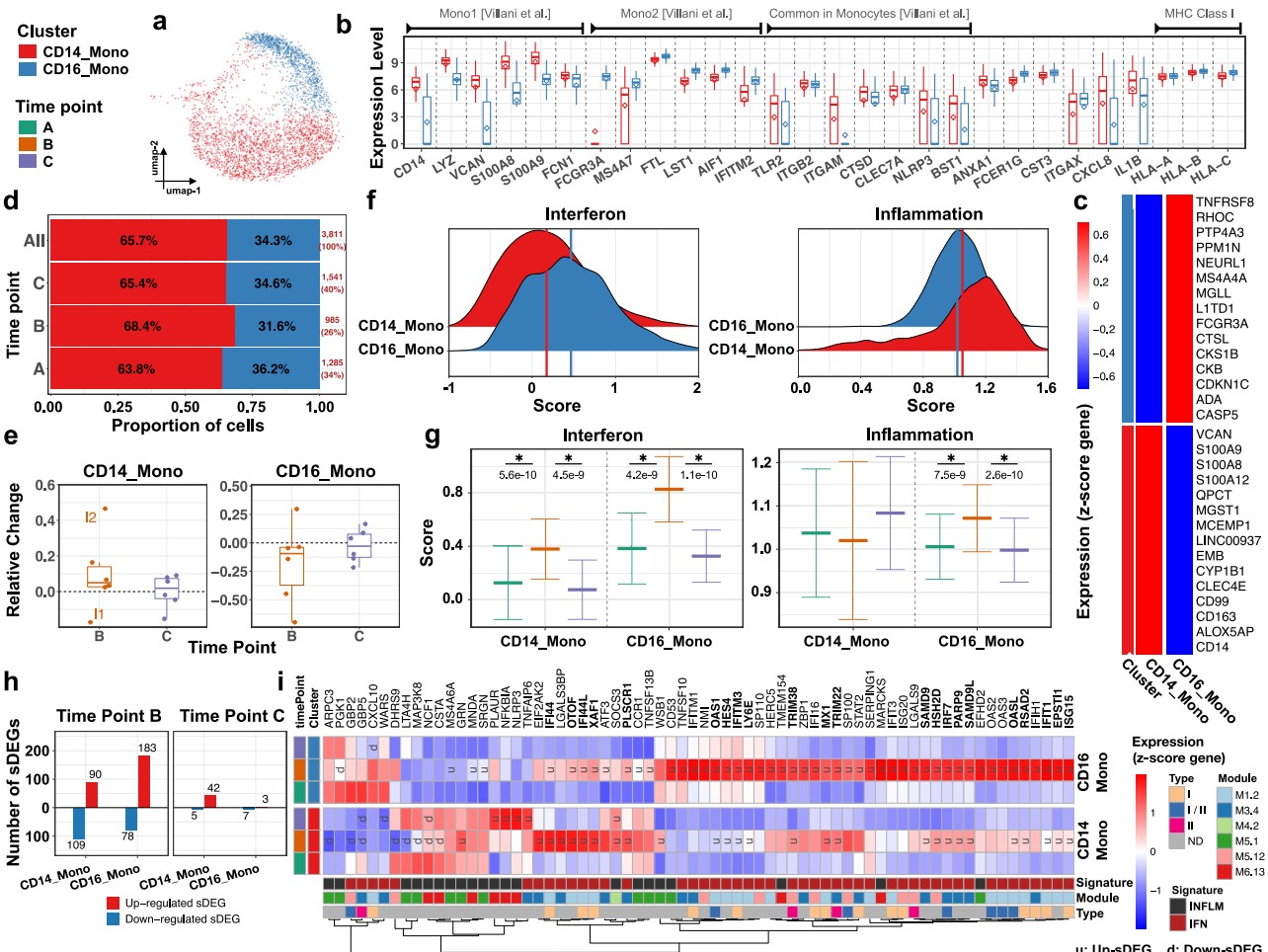

**Fig. 6 | Vaccination induces heightened interferon and inflammatory activity in infants' monocytes.** Subcluster (SC) and time point color codes are shown in the top-left. **a** UMAP plot representing *n* = 2 color-coded SCs encompassing 3811 cells. **b** Expression levels of selected genes for each SC. The results are depicted in boxplots for all cells, in which the upper and lower bounds represent the 75% and 25% percentiles, respectively. The center bars indicate the medians, and the whiskers denote values up to 1.5 interquartile ranges above 75% or below the 25% percentiles. The diamonds indicate the mean. **c** Heatmap representing scaled expression values of the top 15 differentially expressed genes defining each of the SCs. The differentially expressed genes are determined using two-sided Wilcoxon rank-sum tests, with adjusted *p* values less than 0.05. **d** Proportion of cells (labels) per time point for each SC. **e** Relative change of cell abundance at post-vaccination (time points B and C) compared to the baseline (time point A) for each SC. The results are depicted in boxplots, in which the value for each infant (total *n* = 6) is represented by a dot; the upper and lower bounds represent the 75% and 25% percentiles, respectively. The center bars indicate the medians, and the whiskers denote values up to 1.5 interquartile ranges above 75% or below the 25% percentiles.

Data beyond the end of the whiskers are outliers (labeled infants). **f** Interferon score distribution (left panel) and inflammation score distribution (right panel) for each SC ranked from the lowest (top) to highest (bottom). Color-coded vertical lines indicate the average score per SC. **g** Interferon score (left panel) and inflammation score (right panel) calculated for each cell per time point and SC. Error bar indicates standard deviation calculated across cells. Horizontal bar indicates mean. Asterisk (*) indicates a significant difference between two consecutive time points, determined using two-sided *t*-tests, with unadjusted *p* < 0.05 (labels). **h** Number of significantly differentially expressed genes (sDEGs) at post-vaccination (time points B and C) and regulation-type (up-/downregulated) compared to the baseline (time point A) for each SC. **i** Heatmap representing scaled expression values of significantly differentially expressed interferon and inflammatory-related genes per time point for each SC. Genes upregulated at time point B across all SCs are shown in bold. The color keys on the left and bottom represent gene associations. sDEGs are determined as those with a log2-fold-change>0.25 and adjusted *p* < 0.05, calculated using two-sided Wilcoxon rank-sum tests.

and CD16_Mono SCs showed low and comparable IFN score distributions (Fig. 6f, left panel), contrasting with a relatively higher INFLM score in both SCs (Fig. 6f, right panel). IFN scores increased significantly ($p < 0.05$) at time point B in both Mono SCs, (Fig. 6g, left panel) but returned to baseline at time point C. INFLM scores remained unchanged over time in CD14_Mono SC (Fig. 6g, right panel) but showed a significant ($p < 0.05$) increase in CD16_Mono SC at time point B, returning to baseline at time point C (Fig. 6g, right panel). Overall, this analysis offered evidence of innate immune activation involving both IFN and INFLM pathways lasting through at least 7 days post-vaccination in 2-month-old infants.

Differential expression analysis identified a 383 gene vaccine-induced signature (sDEGs: $p < 0.05$, log2-fold-change > 0.25) in both monocyte SCs (Fig. 6h and Supplementary Data 3). The DEGs were predominantly identified at time point B (340/383) (Fig. 6h and Supplementary Fig. S8c). Enrichment analysis confirmed a positive correlation with IFN response modules at this time point B (Supplementary Fig. S8d and Fig. 6i). We also identified 21/383 INFLM-RGs significantly changed at time point B at least in one of the SCs (Fig. 6i). As expected, based on the induction of an IFN-inducible program, we observed an increase in the expression levels of antigen presentation associated genes such as *HLA-A*, *HLA-B*, and *HLA-C*, at time point B in both monocyte SCs (Supplementary Fig. S8e), and a concomitant decrease in the expression levels of *MHCII* genes (Supplementary Fig. S8e). Altogether, vaccination induced significant changes in the blood monocyte compartment of vaccinated infants that include ISGs and INFLM-RGs.

## Vaccination shows natural killer cell phenotypic heterogeneity with changes in interferon-stimulated and cytotoxic-regulated genes

The 3,526 NK cells, which represented the fifth largest immune compartment (5% of PBMCs), yielded two SCs; CD16_NK (3075 cells) and CD56_NK (451 cells) (Fig. 7a and Supplementary Fig. S9a), that displayed the frequencies and phenotypic markers of the known blood NK populations, *KLRB1*, *KLRD1*, *KLRF1* (Fig. 7b)[45]. Thus, CD16_NK expressed *FCGR3A* (CD16) and *CX3CR1*, whereas CD56_NK upregulated *XCL1*, *XCL2*, *CXCR3*, and *NCAM1* (CD56bright NK cells) (Fig. 7b, c). Of note, CD56_NK SC was slightly expanded at time point B (Fig. 7d) in 5/6 infants (Fig. 7e and Supplementary Fig. S9b).

IFN and cytotoxic scores (based on the expression of 137 ISGs and 105 cytotoxic-RGs) revealed that both SCs showed comparable IFN score distribution, while the cytotoxic score was higher in CD16_NK (Fig. 7f). A significant ($p < 0.05$) increase in IFN score at time point B was observed in both NK SCs (Fig. 7g, left panel) and returned to the baseline at time point C. A significant decreasing trend ($p < 0.05$) in cytotoxic gene expression was observed at time point B in both SCs, returning to baseline at time point C (Fig. 7g, right panel).

DE analysis identified a vaccine-induced signature comprising 338 genes (sDEGs: $p < 0.05$, log2-fold-change > 0.25) in both NK SCs (Fig. 7h and Supplementary Data 3). This signature predominantly appeared at time point B post-vaccination (331/338) in the CD16_NK SC (Fig. 7h and Supplementary Fig. S9c). Enrichment analysis (see Method) revealed a positive correlation with interferon response modules at time point B (Supplementary Fig. S9d and Fig. 7i). Thus, 2-month-old infants displayed two major blood NK cell SCs that responded with a significant increase in the expression level of ISGs and decrease in the expression level of cytotoxic-RGs at week 1, both of which returned to baseline levels 1 month after vaccination.

## Vaccination results in early contraction and subsequent expansion of plasmacytoid and conventional dendritic cells

We identified the two main types of DCs: plasmacytoid dendritic cells (pDCs; $n = 117$) and conventional (or classical) dendritic cells (cDCs; $n = 92$), comprising small portions of the data set, 0.17% and

0.13%, respectively. pDCs transcribed well-known markers, including *IRF8*, *IL3RA*, *LILRA4*, *PLD4*, and *ITM2C* (Supplementary Fig. S10a)[46]. Bona fide markers of cDC2s and AXL DCs were identified as well, including *CD1C*, *CLEC10A*, *AXL*, and *PPP1R14A*, while cDC1 markers (*CLEC9A*, *XCR1*)[46] were hardly detected (Supplementary Fig. S10b). While the frequency of both pDCs and cDCs decreased at time point B, a significant ($p < 0.05$) increase in IFN scores was found at this time point, especially in pDCs (Supplementary Fig. S10c–f). ISG scores returned to baseline at time point C when both DC subsets expanded. Thus, at time point B, 2-month-old infants display a contraction of blood pDCs and cDCs, which reverses at time point C and surpasses the baseline.

## Gene two-dimension analysis of IFN response in vaccinated infants reveals distinct gene overexpression in specific cell types

Gene expression analysis is limited by one-dimension, which is the change in the expression level of a given gene across conditions. However, changes in the abundance of cells that express the given genes also provide relevant information. Therefore, to increase our power to identify vaccine-induced signatures, we developed a systematic approach at the individual gene level, which considers both criteria, expression level and cell abundance concurrently. We coined this approach as G2DA (gene two-dimension augmentation; see Methods for details) analysis. We further developed a visualization method, coined here as stream-plot, for monitoring the dynamics of the identified 2d-augmented genes over time (see Supplementary Fig. S11).

We applied G2DA analysis using 137 ISGs and confirmed a vaccine-induced ISG signature that included 79 genes (2d-augmented genes) over all SCs. Multiple points were noted from this analysis (Fig. 8a). While we upheld the G2DA requirements on both time points post-vaccination evenly, the fractional abundance of cells expressing the 2d-augmented genes was intensified at time point B. Interestingly, this signal returned to baseline (pre-vaccination) at time point C in all SCs except plasma cells. The number of identified 2d-augmented genes varied among cell types (Fig. 8b). A substantial fraction (60%) was identified amongst B cells (35%) and CD4+ T cells (25%). NK cells, CD8+ T cells, pDCs, and Monocytes contributed less than 10% each: 9.7%, 9.4%, 7.4%, and 7.1%, respectively. PCs, cDCs, and Eryt cells contributed less than 5% each: 3%, 2.5%, and 0.7%, respectively. No 2d-augmented genes were identified amongst the Mgk cell cluster. Finally, augmentation of some genes was exclusive to specific cell types: *ZBP1*, *MT2A*, *PML*, *CHMP5*, *UNC93B1*, *ABCA1*, *FBXO6*, *GADD45B*, *RHBDF2*, *TRIM56*, and *ZC3HAV1* were specific to B cells; *CASP1*, *DHX58*, *IFIT5*, and *LAMP3* to CD4+ T cells; *IFI16* to CD8+ T cells; *TNFAIP6*, *SERPING1*, *TRAFD1* to Monocytes; *SP110*, *TIMM10*, and *TRIM38* to PCs; and *LY6E* and *REC8* to pDCs. No transcript was identified as 2d-augmented across all cell types (Fig. 8b).

To assess the specificity of the 2d-augmented ISGs amongst cell types, we calculated the average expression of the 79 identified genes within every SC and divided them into 9 groups (G1-G9) by hierarchical clustering (Supplementary Fig. S12). The largest group G1 (23/79) comprised genes that were highly expressed at time point B in CD4+ T cells (CD4_TISGhi SC), B cells (BISGhi SC), and Monocytes (CD16_Mono SC). The second largest group G2 (18/79) comprised genes preferentially expressed in B cells. G3 (12/79) was preferentially expressed in Monocytes. G4 (9/79) was preferentially expressed in CD4+ T cells (CD4_TISGhi, CD4_TMAIT, and CD4_TREG SCs), CD8+ T cells (CD8_TISGhi and CD8_TGdT SCs), and NK cells. G5 (8/79) was preferentially expressed in pDCs. G6 (3/79) was preferentially expressed in B cells and PCs. The three smallest groups did not yield a clear correlation, but overall, G7 (2/79) was preferentially expressed in Mgks, G8 (2/79) showed biased expression in cDCs, and G9 (2/79) was preferentially upregulated in CD8_TISGhi and CD16_NK SCs. Thus, G2DA analysis highlighted multiple cell type-

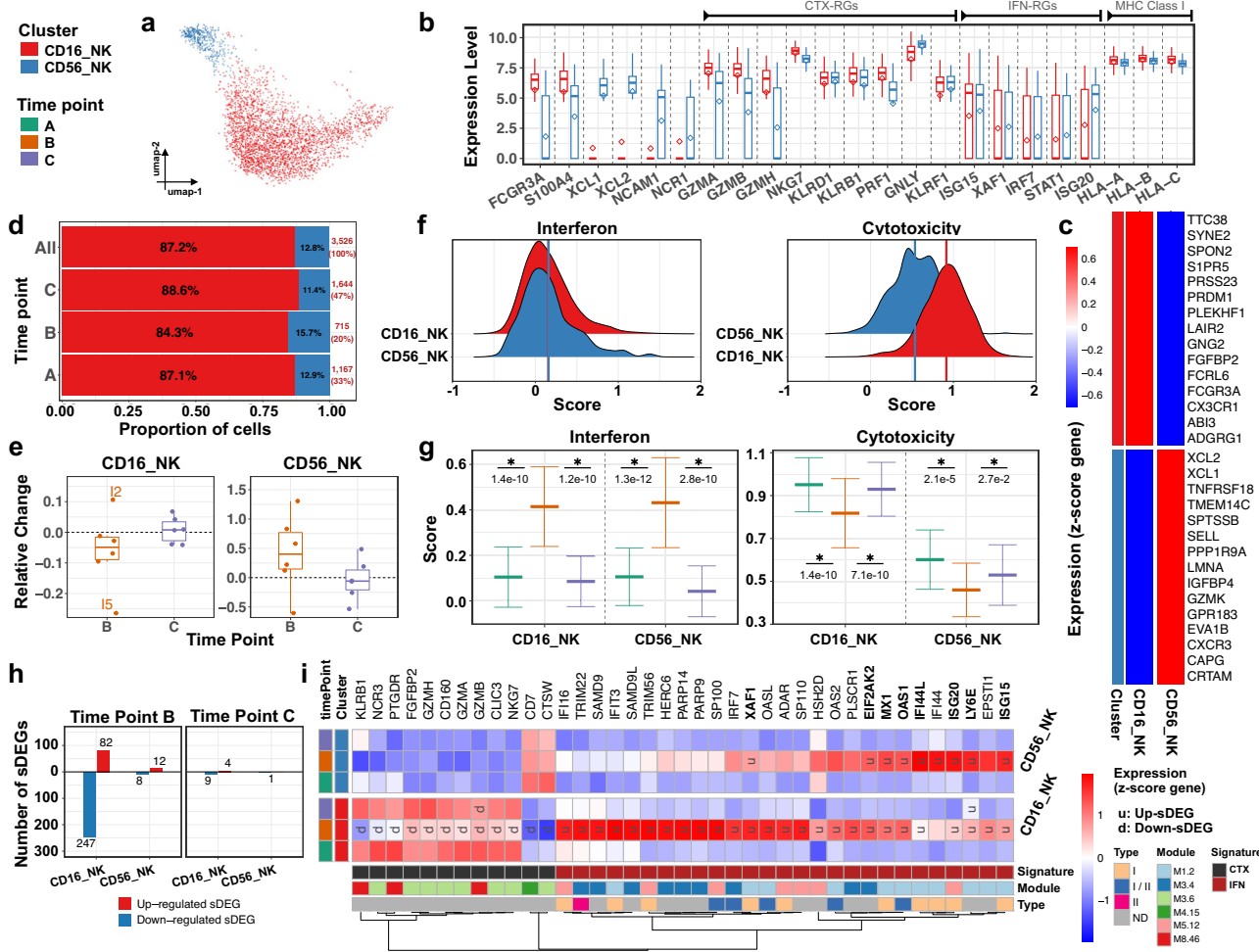

**Fig. 7 | Infants' CD16+ and CD56+ NK cells respond with a significant increase in the expression level of ISGs 1 week post-vaccination.** Subcluster (SC) and time point color codes are shown in the top-left. **a** UMAP plot representing *n* = 2 color-coded SCs encompassing 3526 cells. **b** Expression levels of selected genes for each SC. The results are depicted in boxplots for all cells, in which the upper and lower bounds represent the 75% and 25% percentiles, respectively. The center bars indicate the medians, and the whiskers denote values up to 1.5 interquartile ranges above 75% or below the 25% percentiles. The diamonds indicate the mean. **c** Heatmap representing scaled expression values of the top 15 differentially expressed genes defining each of the SCs. The differentially expressed genes are determined using two-sided Wilcoxon rank-sum tests, with adjusted *p* values less than 0.05. **d** Proportion of cells (labels) per time point for each SC. **e** Relative change of cell abundance at post-vaccination (time points B and C) compared to the baseline (time point A) for each SC. The results are depicted in boxplots, in which the value for each infant (total *n* = 6) is represented by a dot; the upper and lower bounds represent the 75% and 25% percentiles, respectively. The center bars indicate the medians, and the whiskers denote values up to 1.5 interquartile ranges

above 75% or below the 25% percentiles. Data beyond the end of the whiskers are outliers (labeled infants). **f** Interferon score distribution (left panel) and cytotoxicity score distribution (right panel) for each SC ranked from the lowest (top) to highest (bottom). Color-coded vertical lines indicate the average score per SC. **g** Interferon score (left panel) and cytotoxicity score (right panel) calculated for each cell per time point and SC. Error bar indicates standard deviation calculated across cells. Horizontal bar indicates mean. Asterisk (*) indicates a significant difference between two consecutive time points, determined using two-sided *t*-tests, with unadjusted *p* < 0.05 (labels). **h** Number of significantly differentially expressed genes (sDEGs) at post-vaccination (time points B and C) and regulation-type (up-/downregulated) compared to the baseline (time point A) for each SC. **i** Heatmap representing scaled expression values of significantly differentially expressed interferon and cytotoxicity-related genes per time point for each SC. Genes upregulated at time point B across all SCs are shown in bold. The color keys on the left and bottom represent gene associations. sDEGs are determined as those with a log2-fold-change >0.25 and adjusted *p* < 0.05, calculated using two-sided Wilcoxon rank-sum tests.

specific vaccine-stimulated ISG sub-groups from scRNA-seq data in 2-month-old infants, and their normalization at time point C relative to baseline in all cell types except Plasmablasts/Plasma cells.

### Vaccine antibody responses are associated with baseline cell frequencies and day 7 IFN scores

We next proceeded to investigate the association between vaccine responses and cell frequencies in each infant. Our analysis revealed that infants with the Best-R (I6) response exhibited higher frequencies of plasma cells and monocytes, both at baseline and on day 7 (Fig. 9a). In addition, the Best-R (I6) and Good-Rs (I1 and I2) showed, in general, higher B cell frequencies. The dominance was

emboldened at time point A and C, and to lesser extent at time point B across B SCs. Subsequently, we examined the relationship between vaccine responses and IFN scores in each infant. Our analysis demonstrated that infants with the Best-R (I6) response, as well as the Good-Rs (I1 and I2), exhibited significantly higher IFN scores at time point B across various immune cell types. (Fig. 9b). There was no significant relationship between baseline IFN scores and antibody responses.

Thus, higher frequencies of plasma cells, B cells and monocytes at baseline together with the capacity to induce an ISG program around day 7 post-vaccination were associated with the most robust antibody responses among 2-month-old infants.

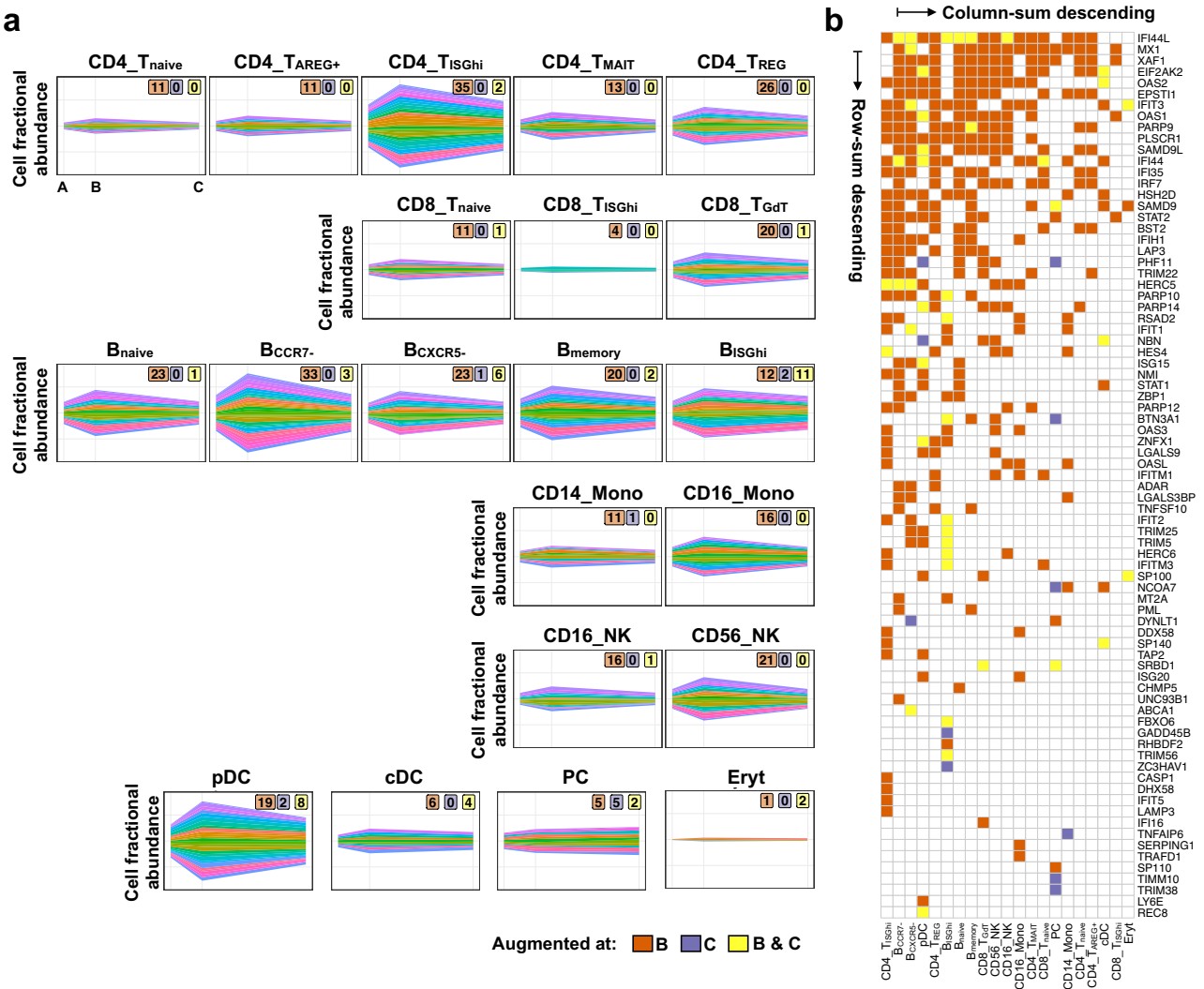

**Fig. 8 | Universal heightened interferon activity in vaccinated infants 1 week post-vaccination. a** Fractional abundance of cells (*y*-axis) expressing 2D-augmented genes (color-coded ribbons) at each time point (*x*-axis) for each Sub-cluster (SC). Labels at the top-right of each panel represent the number of 2D-augmented genes at time point B (orange), time point C (purple), or both time points B and C (yellow). The *y*-axis scale is fixed across all panels. Refer to Supplementary Fig. S9 for a detailed description of ribbon plots. **b** A binary color-coded map representing 2D-augmented genes (*n* = 79 rows) across SCs (*n* = 21 columns). Colors indicate augmentation occurrence at time point B (orange), C (purple), or both B and C (yellow). Rows are ranked top-to-bottom descending based on the number of SCs that a given gene was found to be 2D-augmented. Columns are ranked left-to-right descending based on the number of 2D-augmented genes per each SC.

## Discussion

Considerable progress has been made in the systems immunology of vaccination of adults, but the field lags considerably behind when it comes to infants due, in part, to challenges in enrolling and obtaining longitudinal blood samples at this early age. This gap also has important implications, as infants suffer from considerable infection-related mortality that could be significantly reduced by optimizing vaccine strategies[4]. The study of infants offers however a window of opportunity to understand primary immune responses in the context of a naïve, developing immune system. Our study represents a longitudinal evaluation of the transcriptional response in 2-month-old infants following the administration of the first set of routine vaccines. This evaluation was achieved by combining bulk transcriptome analyses in a first cohort followed by a second higher resolution analysis using single-cell RNA-seq in an independent cohort of infants. The transcriptional analyses were combined with serologic analyses of responses to multiple vaccine antigens, although with slight variation between cohorts because the limited sample volumes available. As a result, our study provides comprehensive insight into the baseline status of both whole blood and peripheral immune cell types and their transcriptional responses to vaccination.

We found that the overall titers of maternal antibodies measured at baseline before immunization were low in most infants, except for tetanus (in both cohorts) and pertussis responses (only analyzed in the first cohort) most likely reflecting vaccination with Tdap during pregnancy. Thus, it is unlikely that vertical antibody transmission had a significant effect in modulating the observed responses to the 2-month vaccines. Hepatitis B antibody titers at baseline (measured in the second cohort), obtained after the birth dose, were also low but increased uniformly post-revaccination, possibly as the result of the vaccine booster effect and memory build-up. Priming of the immune system with the remaining vaccines induced limited and markedly heterogeneous antibody responses, highlighting significant individual variation already at this early age. The substantial individual variability and overall limited antibody responses observed further support the rationale for administering three or four vaccine doses recommended in the majority of infant vaccination programs.

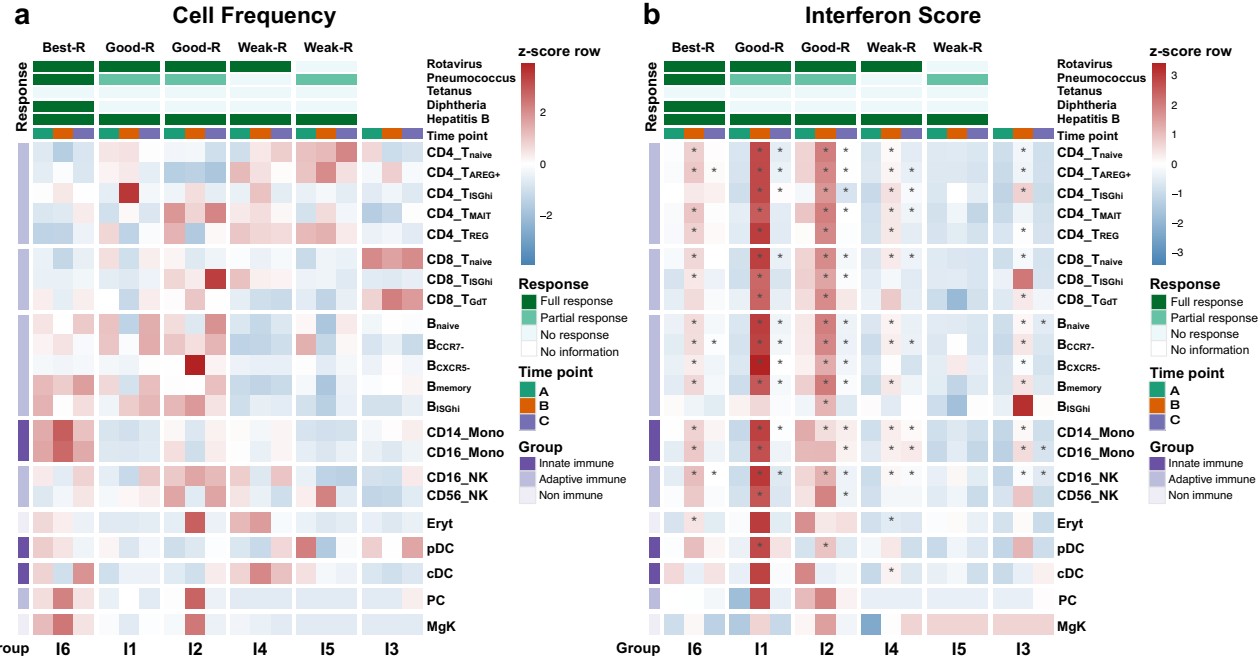

**Fig. 9 | Antibody response in 2-month-old infants from single-cell cohort.** **a** Heatmap representing the proportion of cells per time point and subcluster (SC) for each individual infant. The color key on the left represents cell-type immune associations, and the color key on the top represents the infant's serologic response associations. **b** Heatmap representing the average interferon score per time point and SC for each individual infant. The color key on the left represents cell-type immune associations, and the color key on the top represents the infant's serologic response associations. An asterisk (*) indicates a significant difference of $p < 0.05$, calculated by a two-sided $t$-test comparing post-vaccination (time points B and C) interferon scores versus the baseline (time point A).

In addition to detecting antibody titers, our study profiled the blood immune transcriptome using whole blood bulk transcriptome followed by more granular scRNA-seq analysis in two independent infant cohorts. The longitudinal analysis included baseline and two time points post-vaccination. Deconvolution of the whole blood transcriptome using a modular analytical tool to the initial cohort allowed us to capture transcriptional changes in genes related to major immune pathways and cell populations. At day 7 post-vaccination, we observed a substantial increase in the expression of three IFN-related gene modules, a monocyte gene module, and several modules associated with inflammation. Also, on day 7 the bulk transcriptome analysis showed mild underexpression of T cell genes, likely reflecting the marked difference with the prominent overexpression of interferon and inflammation genes present at that time. Similar observations have been previously reported in transcriptome analysis performed in children with acute viral infections[47]. This potential limitation of the bulk transcriptome can now be addressed with the higher resolution provided by the single-cell analysis.

The analysis of the second cohort at the single-cell level provided insight into the responses of this young population. As reported in adults[25], baseline frequencies of B cells and plasma cells were higher in the best responders. In addition, day 7 monocyte frequencies and IFN scores were also associated with infant's antibody responses. Overall, these results underscore the importance of a coordinated effort between the adaptive and innate arms of the immune system for optimal response to vaccines early in life.

As expected, the infant lymphocyte compartment was predominantly naive at baseline. However, memory B cells and PCs displaying switched isotypes were detected, supporting ongoing isotype switching at the time Tfh cells (CXCR5+CD4+ T cells) were not readily detected in the blood. Interestingly, a CXCR5neg naive B cell population expressing markers associated with "extra-follicular" differentiation[43] was present at baseline and uniquely expanded post-vaccination. Thus, extra-follicular pathways might contribute to vaccine responses and

perhaps account for the need for repeated vaccine doses to induce long-term memory responses at this early age.

Within the CD4+ T cell compartment, Th1 and Th17 cell markers were not identified, while cells expressing *GATA-3* and *STAT6* were readily detected. Additional T cell subsets included regulatory T cells, which have been previously described in the fetal and infant immune system as major contributors to immune tolerance[48,49]. Interestingly, we observed a wide expression of cytotoxic transcripts, including *GZMM* and *GZMA*, within the infant's CD4+ T cells, which contrasts with the more limited expression of these markers within memory CD4+ populations in older age groups.

Administration of the combined vaccines resulted in significant transcriptional changes in CD8+ T cells that returned to baseline at week 4 post-vaccination. At this late time point, however, we observed an expansion of an ISGhi and cytotoxichi TEMRA-like population, which surprisingly was already present at baseline. Whether this population is preferentially elicited by specific vaccines, including those containing live virus, remains to be determined. Previous transcriptional studies in adults immunized with Yellow Fever vaccine (a live attenuated virus) reported overexpression of *KIR2DL3*, *PRF1*, *GNLY*, and *GZMB* (defined as NK genes) at day 7 after the vaccine. However, whether these cytotoxic genes remained overexpressed at later time points was not reported[21,23,50].

Discrete cell subclusters expressing high ISG levels, especially within naive CD4+ T and B lymphocytes, were detected prior to vaccination using the single-cell analysis. Baseline ISGhi cell populations might reflect a tonic activation of the IFN pathway[51]. These cells are also found in older healthy children and are expanded in pediatric-onset systemic lupus erythematosus (SLE) patients[41]. The actual meaning/function of these ISGhi cells remains to be understood. At day 7–8 post-vaccination, transient signatures were induced within each immune cell type analyzed. The most prominent consisted of ISG upregulation that spanned across most cell clusters and subclusters. Marked ISG overexpression was observed at day 7 in the bulk analysis of the initial

cohort, but the extent and the frequency of cell populations involved in this response was impressive at the single-cell level. Indeed, the magnitude, duration, and extent of cell populations involved in the ISG response were higher than those reported previously in adult Influenza vaccine studies. In those studies, ISGs were mostly identified at day 1–3 after post-vaccination[22,24,25,52]. The prolonged infant ISG response may be due to the simultaneous administration of multiple vaccines, including live viruses, and/or the broad repertoire of naive cells, which might be more prone to activation in these young infants.

Although not as prominent as ISGs, bulk transcriptome analyses also identified early overexpression of inflammation-related genes, as previously observed with pneumococcus polysaccharide vaccine[24]. Single-cell analysis confirmed these findings (i.e., increased expression of *CCR1* and *TNFSF13B* within M5.1), specifically in CD16 monocytes.

Finally, the bulk transcriptome analysis showed increased expression of modules corresponding to B cell and plasma cell genes at 4 weeks post-vaccination, and the scRNA-seq analysis confirmed an increased frequency of $B_{naive}$ and $B_{memory}$ cells suggesting an expansion and/or more persisting activation of the B cell compartment. It is tempting to associate these changes with the effect of the multiple vaccines administered, but age-related changes could have also contributed.

There are limitations in this study. First, our study protocol did not permit us to collect samples within the first 3 days after vaccine administration, a critical time for assessing innate immune responses. Second, the number of enrolled infants in the single-cell analyses ($n = 6$) confines the generalizability of our findings. Third, the single-cell study is limited in detecting cell types sensitive to cryopreservation protocols such as neutrophils, which represent an important population in infants' response to infection[8]. The bulk transcriptome analysis captures however neutrophil gene expression changes, although with different level of resolution. Fourth, the low number of certain cell populations such as PCs and DCs confined their analyses, although significant vaccine-related changes could be observed in these compartments. Finally, our study reports a cumulative immune response derived from the simultaneous administration of multiple vaccines, as is required in this age group, and did not allow us to examine vaccine-specific immune cells. Therefore, any attempt to attribute the identified signatures to particular vaccines remains speculative. The significant variation observed in both antibody and immune cell responses highlights the need for future studies focused on young infants. Furthermore, it could argue for the development of more personalized vaccination strategies for this population.

Despite these limitations, the study presents a unique longitudinal and systems-level analysis of the immune system in vaccinated young infants by combining two independent cohorts with different levels of resolution. This approach, conducted within the context of vaccination, offers insight into the steady-state as well as the dynamic nature of the transcriptome of peripheral immune cells and antibody responses upon antigenic challenge in early life.

## Methods

### Study design
This study was approved by the Nationwide Children's Hospital (NCH) IRB (18-00591 and 10-00028) where both cohorts were recruited and enrolled. Written informed consent was obtained from all children's guardians before study participation, and consent was signed by all parents /guardians prior to all study procedures. Blood samples (between 2.1 and 5.3 mL) were obtained by venipuncture. To reduce the stress and pain of the participating infants during the blood draws required for the study, infants were swaddled, and we offered them an oral sucrose solution such as sweet-ease. All infants were previously asymptomatic and did not receive antibiotics, steroids, or any other medications in the 2 weeks prior to vaccination. Further clinical details of all participant infants are summarized in Supplementary Data 1.

### Antibody assays
For the initial cohort, samples ($n = 19$ samples for serology) were analyzed at the Center for Infectious Disease Control, National Institute for Public Health and the Environment (RIVM), Bilthoven, The Netherlands. Antibodies were determined using established multiplex ELISA assays against (1) conjugated polysaccharide *Streptococcus pneumoniae* vaccine (PCV13), including 13 different serotypes: Ps1, Ps3, Ps4, Ps5, Ps6a, Ps6b, Ps7f, Ps9v, Ps14, Ps18c, Ps19a, Ps19f, Ps23f; (2) *Haemophilus influenzae* vaccine (Hib); (3) diphtheria, tetanus, and pertussis vaccine (DTaP), that includes Ptx (Pertussis toxin), FHA (Filamentous hemagglutinin) and Prn (Pertactin) for *Bordetella pertussis*, Diphteria toxin (DTxd) for *Corynebacterium diphtheria* and Tetatus toxin (Ttx) for *Clostridium tetani*. Cut-off levels of antibodies for seropositivity were >0.35 μg/ml for PCV13; >0.15 μg/ml for Hib; 25 EU/ml for pertussis antibodies and 0.01 IU/ml for DTxd and Ttx. For the second infant cohort ($n = 6$) serum IgG antibodies were measured in the Nationwide Children's Hospital clinical laboratory using standard assays for hepatitis B (Chemiluminescent microparticle immunoassay[53], diphtheria (EIA), tetanus (https://www.bindingsite.com/en/our-products/immunoassays/plasma-screening/vaccine-response?disclaimer=1), and quantitative multiplex bead immunoassays for Haemophilus influenzae type b, and pneumococcus. Serum anti-rotavirus IgA antibody titers were measured by EIA using the 8912-virus lysate at Cincinnati Children's Hospital Medical Center Translational Laboratory[54].

### Bulk transcriptome preparation and analysis
Whole blood samples (1 mL) for transcriptome analyses were collected in Tempus tubes (Applied Biosystems, CA, USA) and stored at −20 °C. After processing, RNA was hybridized into Illumina Human HT-12 v4 beadchips (47,323 probes) and scanned on the Illumina Beadstation 500[47,55,56]. Illumina GenomeStudio software was used for pre-processing the data (background subtraction and normalizations). Downstream analysis was performed in R environment. Differentially expressed genes between the groups were identified using limma package with adjusted p value less than 0.05 and fold change greater than 1.5. To assess the immune function of the differentially expressed genes between the groups we used a modular transcriptional repertoire framework[32,33].

### Sample processing and blood preparation for scRNA-seq
The freezing medium was 10% DMSO + 90% FBS. PBMCs were thawed quickly at 37 °C and into DMEM supplemented with 10% FBS. Cells exhibiting a viability rate of less than 70% were excluded. Cells were quickly spun down at 400 g, for 10 min. Cells were washed once with 1 × PBS supplemented with 0.04% BSA and finally re-suspended in 1 × PBS with 0.04% BSA. Viability was determined using trypan blue staining and measured on a Countess FLII. Briefly, 12,000 cells were loaded for capture onto the Chromium System using the v3 single-cell reagent kit (10X Genomics). Following capture and lysis, cDNA was synthesized and amplified (12 cycles) as per the manufacturer's protocol (10X Genomics). The amplified cDNA was used to construct an Illumina sequencing library and sequenced on a single lane of a HiSeq 4000. All three time point samples from each individual child were run within the same batch.

### Single-cell raw data processing
Illumina basecall files (BCL) were converted to fastqs using cellranger v3.0.2, which uses bcl2fastq v2.17.1.14. FASTQ files were then aligned to the hg19 genome and transcriptome using the cellranger v3.0.2 pipeline, which generates a gene–cell expression matrix.

### Single-cell gene expression analysis
10× cellranger count matrices of PBMCs from 18 samples (6 infants and 3 sampling time points) were retrieved. Next, the Scrublet python

package (version 0.2.2)[57] was used to predict and remove cells with a doublet score larger than 0.25 from each sample. Then, quality control was performed on the gene expression matrices as follows: (1) genes that were not detected in at least 3 cells were discarded; (2) cells with fewer than 750 total unique transcripts were removed; (3) cells displaying a unique gene count of less than 250 and larger than 3500 were considered outliers and discarded; and (4) cells in which larger than 15% of the transcripts mapped to the mitochondrial genes were filtered out.

Filtered gene expression matrices were merged and processed in accordance with the standard Seurat R workflow (version 4.0.3)[36]. We used the NormalizeData function to normalize the total number of reads in each individual cell to count-per-million (CPM). Next, the FindVariableFeatures function was used to select the 3000 genes with the highest variance using the *vst* (variance stabilizing transformation) method. Then, the data were regressed against the percentage of mitochondrial genes and scaled to unit variance using the ScaleData function. Principal component analysis (PCA) was performed using the RunPCA function, followed by the RunHarmony function, from Harmony R package (version 1.0), to correct batch effects across samples[58].

The first 40 Principal Components (PCs) were used in the Find-Neighbors algorithm to construct the nearest-neighbor graph. Next, the Louvain modularity optimization algorithm in the FindClusters function was used to generate the clusters, while the resolution was set to 0.8. Then, the RunUMAP function was used to perform uniform manifold approximation and projection (UMAP). Finally, multiple rounds of marker identification, cell type annotation, and manual inspection and doublet removal, were performed to create the final UMAP. A summary of the pipeline and results can be found in Supplementary Fig. 2a–e.

### Functions used for downstream analysis

The FindMarkers function was used to identify differentially expressed genes (DEGs) by the Wilcoxon rank sum test. Regulations with absolute log2-fold change larger than 0.25 and adjusted *p* value less than 0.05 were called significant. The Seurat function FindAllMarkers was used to find markers for a specific cluster against all remaining cells. The Seurat function DotPlot was used to visualize the gene expression with a dot plot. The Seurat function DimPlot was used to visualize the final UMAP on a 2D scatter plot. The Seurat function AddModuleScore was used to calculate interferon, inflammation, and cytotoxicity score for each individual cell based on the average expression of associated gene lists. A positive score would suggest that this module of genes is expressed in a particular cell more highly than would be expected, given the average expression of this module across the population. Enrichment between modules and sDEGs was analyzed using the GeneOverlap R package (version 1.26.0). GeneOverlap utilizes Fisher exact tests to determine whether enrichment is significant ($p < 0.05$) and reports odds ratio and Jaccard index to denote the level of enrichment[59]. The default arguments are used in all functions except those mentioned.

### Gene 2D augmentation analysis

A gene is considered 2D-augmented if: (1) at least 10 cells express the given gene post-vaccination; (2) a minimum of 50 reads[16–18] are observed on average amongst cells expressing the given gene post-vaccination; (3) cells expressing the given gene comprise at least 1% of the cells at time point B or C; (4) an absolute fold increase of at least 1.5 in fractional abundance is observed at time point B or C relative to vaccination-day; (5) an absolute fold increase of at least 1.5 in expression level is observed at time point B or C relative to vaccination-day. The fractional abundance is defined as the number of cells expressing a given gene divided by the total number of cells observed at a given time point. Thresholds used in (1), (2), and (3) were chosen as low as

possible to filter out low expressed genes while maintaining a reasonable baseline for comparison. Thresholds used in conditions (4) and (5) were chosen amongst common values from literature (see Supplementary Fig. 11).

### Reporting summary

Further information on research design is available in the Nature Portfolio Reporting Summary linked to this article.

## Data availability

The single-cell RNA count matrices generated in this study have been deposited in the GEO database under accession code GSE204716. FASTQ files are available in the dbGAP database under accession code phs002926.v1.p1. The Bulk RNA data generated in this study have been deposited in the GEO database under accession code GSE237318. Source Data are provided with this paper.

## Code availability

The code used for the analysis is available on GitHub: https://github.com/nourin-nn/infants_vaccines_cocktail (https://zenodo.org/records/10049360).

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

## Acknowledgements

We are grateful to the infants and their families for their participation in the study. Special thanks to Monica McNeal and David Bernstein, Cincinnati Children's Hospital Medical Center for their assistance with the rotavirus antibody assays, helpful discussions and their review of the manuscript. We thank the Nationwide Children's Hospital Infectious

Diseases clinical research team for study subject recruitment and management; M. Collet and L. Choquette for help with IRB and data upload to dbGAP. We thank JAX single-cell core facility for their help with single-cell assays, JAX genomic technologies for their help with generating the sequencing data, and JAX Research IT for help with data management. The study was funded by NIH grants AI131386 to O.R. and J.B. and AI168632 to O.R. and V.P.

## Author contributions

N.N. performed analyses. D.N.-B. designed the experiments and processed data. E.B., R.G.-C., B.S., S.H., R.M., S.M., and A.M. enrolled patients, collected, processed, and analyzed samples and patient information. A.L. G.S. and F.V.K. performed laboratory assays and interpreted data. Z.X. analyzed and interpreted bioinformatic data. A.M. interpreted assay data and integrated with patient information. N.N., D.N.-B., J.B., V.P., and O.R. jointly interpreted the data and wrote the manuscript. All authors reviewed and contributed to the final manuscript. J.B., V.P. and O.R. jointly directed the study.

## Competing interests

While this study was performed, J.B. served on the Board of Directors (BOD) for Neovacs, is a BOD member and stockholder for Ascend Biopharmaceuticals, Scientific Advisory Board (SAB) member for Cue Biopharma, and stockholder for Sanofi. J.B. is currently an employee of Immunai and holds shares from Immunai, Merck and Novartis. N.N. currently holds the position of principal scientist at Sanofi. A.M. has received research grants from Janssen and Merck, fees for participation in advisory boards from Janssen, Merck, Sanofi-Pasteur, and fees for lectures from Sanofi-Pasteur and Astra-Zeneca. O.R. has received research grants from the Bill & Melinda Gates Foundation, Merck and Janssen; and fees for participation in advisory boards from Merck, Sanofi-Pasteur, Lilly, Pfizer and Adagio; and fees for lectures from Pfizer, Astra-Zeneca, and Sanofi-Pasteur. None of these fees were related to the research described in this manuscript. The remaining authors declare no competing interests.
