## [Peer review file · Nature Communications]

REVIEWER COMMENTS

Reviewer #1 (expert in transcriptomics):

The manuscript has significantly been improved and it is now easier to follow. Here are my outstanding comments.

- Why was the description of PCs grouped with that of DCs?

"Thus, 2-month-old infants display a contraction of blood pDCs and cDCs and an expansion of PCs expressing ISGs together with skin and intestine homing receptors at day 7 post-vaccination. Subsequently, at time point C, PC numbers returned to baseline while DCs increased."

This description gives the impression that the changes in the differential abundance of DCs and PCs are somehow mechanistically related without any additional explanation. Would it make more sense if the description of PCs is grouped with (or proximal to) that of B cells, as the implication for changes in the differential abundance in certain subsets of B cells and PCs might make more sense mechanistically?

- I expect some discussion on multiple cell type-specific vaccine-stimulated ISG sub-groups. Are the findings at least partially known or if they are novel? What could be the biological/mechanistic significance of the findings? Are they children-specific?

- In Figure 8, would it make more sense to group and illustrate the samples in the order (or reverse order) of Best-R, Good-Rs, and Weak-R instead of I1-I6? In addition, the Best/Good/Weak response group assignment could be directly labeled in the figure so the readers could easily spot the trends.

- I still have some doubts regarding the use of the word "correlation" explained in Figure 8.

The authors suggested that "Correlation analyses of vaccine responses IFN scores revealed that the Best-R (I6) and Good-Rs (I1 and I2) had higher IFN scores at time point B for most immune cell types (Figure 8B). There was no significant correlation between baseline IFN scores and antibody responses" in the main text and "despite the limited number of samples we were able to identify significant associations" in the rebuttal letter.

I understand that, as mentioned in the rebuttal letter, the authors "chose to analyze 'correlations', as they mean an association and do not imply cause-effect or a mechanistic implication." However, correlation analyses and the use of the word "significant" imply the use of a statistical method (e.g. Pearson, Spearman, or Kendall) to measure the strength of the association. Was such a statistical test performed? If so, please include it in the Method section. If not and the "correlation" was done by just spotting the trends, please be explicit about it and try not to avoid making an impression that a correlation test was done. This is what I meant when I said you could soften the claim and be more careful with the word "correlation", which has a special meaning in the statistical context.

Reviewer #2 (expert in infectious disease pediatrics):

The revised paper is improved. Discussing the serological data for 5 of 6 infants in revised figure 1 makes the key point more accessible. As with the original paper, the authors should be commented for addressing this issue in infants, especially given the limited quantity of samples available for this age group. Some suggestions for further improvement, clarification.

1. Given the limited number of infants studies, are there statistical analysis that can be used to support the conclusions described in the last sentence of the abstract? If so, these should be incorporated into the Figure 8?
2. The data in S1D and S1E I think are essential for understanding the single cell gene expression profiling. I would consider incorporating this into the main figures.
3. Figure S2B is not labelled? What do the different colors mean? Is this plot showing cells from all 18 samples?
4. Is the lack of baseline antibody titers for antigens other than tetanus expected? Please expand discussion on this point comparing other studies on persistence of vertically transferred IgG to other antigens in a comparable population?
5. Is the heterogeneity in serological response to vaccines at 2 months of age expected? Please expand discussion on this point.

Reviewer #3 (expert in 'Systems' Immunology and inborn errors or immunity):

In the rebuttal and revised manuscript only minor edits were performed and no new additional data was added. Therefor my original concern of shallow sampling, uneven representation of cells from different samples skewing the analyses and the lack of potentially any vaccine specific cells in the results, means that the conclusions are not resting on particularly solid data.

I still think that more data should be added from additional samples and that these samples should be sampled more deeply (more cells/sample sequenced) in order to ensure robustness of these results.

RESPONSE TO REVIEWERS' COMMENTS

Reviewer #1 (expert in transcriptomics):

The manuscript has significantly been improved and it is now easier to follow. Here are my outstanding comments.

We appreciate Reviewer #1 overall assessment that the manuscript is significantly improved.

Comment #1. Why was the description of PCs grouped with that of DCs?

“Thus, 2-month-old infants display a contraction of blood pDCs and cDCs and an expansion of PCs expressing ISGs together with skin and intestine homing receptors at day 7 post-vaccination. Subsequently, at time point C, PC numbers returned to baseline while DCs increased.”

This description gives the impression that the changes in the differential abundance of DCs and PCs are somehow mechanistically related without any additional explanation. Would it make more sense if the description of PCs is grouped with (or proximal to) that of B cells, as the implication for changes in the differential abundance in certain subsets of B cells and PCs might make more sense mechanistically?

“scRNA-seq analysis yielded 55 Plasmablasts/Plasma cells (0.08% of PBMCs). At baseline, infant’s PCs were characterized by the expression of IGHM (IgM), IGHG1 (IgG), and IGHA1 (IgA), together with CD27, PRDM1 (Blimp-1), XBP1, and MZB1 (Figure S7A). At time point B, PC frequencies expanded, and interferon scores increased significantly ($p < 0.05$) (Figure S7B), returning to the baseline at time point C. We observed, however, an increased expression of discrete sets of ISGs at each time point post-vaccination (Figure S7C). Interestingly, while few infant PCs expressed CCR9, a chemokine receptor associated with intestinal migration, up to half of them expressed CCR10, a homing receptor for skin and intestine (Figure S7A). Thus, 2-month-old infants display an expansion of PCs expressing ISGs together with skin and intestine homing receptors at day 7 post-vaccination, returned to baseline at time point C”.

Response #1. We have modified the result section and the discussion to address this point.

Comment #2

I expect some discussion on multiple cell type-specific vaccine-stimulated ISG sub-groups. Are the findings at least partially known or if they are novel? What could be the biological/mechanistic significance of the findings? Are they children-specific?

Response #2. We agree that such robust, reproducible (two independent studies), generalized and prolonged ISG responses across most immune cell populations after vaccination is indeed remarkable and not reported earlier. Encouraged by the reviewer’s comments we have now included a more in-depth discussion of this observation.

“Marked ISG overexpression was observed at day 7 in the bulk analysis of the initial cohort, but the extent and the frequency of cell populations involved in this response was impressive at the single cell

level. Indeed, the magnitude, duration, and extent of cell populations involved in the ISG response were higher than those reported previously in adult Influenza vaccine studies”.

Comment #3

In Figure 8, would it make more sense to group and illustrate the samples in the order (or reverse order) of Best-R, Good-Rs, and Weak-R instead of I1-I6? In addition, the Best/Good/Weak response group assignment could be directly labeled in the figure so the readers could easily spot the trends.

Response #3

In response to the reviewer's request, we have revised Figure 8 (new Figure 9) to display infants in order of their vaccine response, rather than using a predefined sequence. Additionally, we added labels at the top of the panels to indicate the response group assignments.

Comment #4

I still have some doubts regarding the use of the word “correlation” explained in Figure 8.

The authors suggested that “Correlation analyses of vaccine responses IFN scores revealed that the Best-R (I6) and Good-Rs (I1 and I2) had higher IFN scores at time point B for most immune cell types (Figure 8B). There was no significant correlation between baseline IFN scores and antibody responses” in the main text and “despite the limited number of samples we were able to identify significant associations” in the rebuttal letter.

I understand that, as mentioned in the rebuttal letter, the authors “chose to analyze ‘correlations’, as they mean an association and do not imply cause-effect or a mechanistic implication.” However, correlation analyses and the use of the word “significant” imply the use of a statistical method (e.g. Pearson, Spearman, or Kendall) to measure the strength of the association. Was such a statistical test performed? If so, please include it in the Method section. If not and the “correlation” was done by just spotting the trends, please be explicit about it and try not to avoid making an impression that a correlation test was done. This is what I meant when I said you could soften the claim and be more careful with the word “correlation”, which has a special meaning in the statistical context.

Response #4

We thank the reviewer for pointing this out. We have revised that section and now use the word “association”.

Reviewer #2 (expert in infectious disease pediatrics):

The revised paper is improved. Discussing the serological data for 5 of 6 infants in revised figure 1 makes the key point more accessible. As with the original paper, the authors should be commented for addressing this issue in infants, especially given the limited quantity of samples available for this age group. Some suggestions for further improvement, clarification.

We appreciate Reviewer #2 general comment about the improved manuscript and the challenges associated to the conduction of such studies in young healthy infants.

Comment #1

1. Given the limited number of infants studies, are there statistical analysis that can be used to support

the conclusions described in the last sentence of the abstract? If so, these should be incorporated into the Figure 8?

Response #1

Following the reviewer's request, we have conducted a statistical analysis to compare the post-vaccination interferon scores with the baseline. This analysis has been incorporated in Figure 8B (new Figure 9B), and the legend has been modified accordingly. Please note that such analysis cannot be performed on panel A, which presents the absolute values of cell frequency rather than a distribution of values per group.

Comment #2.

The data in S1D and S1E I think are essential for understanding the single cell gene expression profiling. I would consider incorporating this into the main figures.

Response #2

We appreciate the reviewer's comment that those figures provide important data; we would like to include them. We are concerned, however, about the extension of the manuscript (9 main figures, 12 supplementary figures, and 3 supplementary files). We would appreciate the Editor's advice regarding the inclusion of these supplemental data in the main figures of the manuscript.

Comment #3

Figure S2B is not labelled? What do the different colors mean? Is this plot showing cells from all 18 samples?

Response #3

We apologize for the lack of labelling. The plot does actually show cells from all 18 samples and demonstrates the strong overlap among them. Thus, we removed the colors legend as they provided minimal additional information to the plot. We have modified the figure legend accordingly (new Figure S3B).

Comment #4.

Is the lack of baseline antibody titers for antigens other than tetanus expected? Please expand discussion on this point comparing other studies on persistence of vertically transferred IgG to other antigens in a comparable population?

Response #4

Most studies focused on antibodies transferred across the placenta commonly report antibody titers in cord blood samples and not in infants. Following the reviewer comment we are now including additional baseline data at 2 months of age from an independent cohort of 19 infants. As shown in Supplementary Figure 1A, at baseline the majority of infants had low antibody titers against most pneumococcus serotypes, Hemophilus influenzae type b, and diphtheria. On the other hand, they had detectable antibodies against tetanus and pertussis antigens, most likely reflecting vaccination with Tdap during pregnancy.

Comment #5.

Is the heterogeneity in serological response to vaccines at 2 months of age expected? Please expand discussion on this point.

Response #5

Following our response #4, we have incorporated the additional serological data from 19 infants (Supplementary Figure 1A) and expanded our discussion.

We found that the overall titers of maternal antibodies measured at baseline before immunization were low in most infants, except for the tetanus (in both cohorts) and pertussis responses (only analyzed in the first cohort) most likely reflecting vaccination with Tdap during pregnancy. Thus, it is unlikely that vertical antibody transmission had a significant effect in modulating the observed responses to the 2-month vaccines .../... The substantial individual variability and overall limited antibody responses observed further support the rationale for administering three or four vaccine doses recommended in the majority of infant vaccination programs.

Reviewer #3 (expert in 'Systems' Immunology and inborn errors or immunity):

In the rebuttal and revised manuscript only minor edits were performed and no new additional data was added. Therefore my original concern of shallow sampling, uneven representation of cells from different samples skewing the analyses and the lack of potentially any vaccine specific cells in the results, means that the conclusions are not resting on particularly solid data.

I still think that more data should be added from additional samples and that these samples should be sampled more deeply (more cells/sample sequenced) in order to ensure robustness of these results.

Response to Reviewer #3

We appreciate Reviewer #3's overall assessment. Following his/her comments and to ensure the robustness of our findings, we have now included a new set of bulk transcriptomics data generated from 24 infants, 19/24 with serologic responses. Initially, we had decided against incorporating this dataset in our manuscript, as we were concerned the overall complexity of the data would render the manuscript difficult to read. However, thanks to this referee's encouragement, we might have found a way to include this independent cohort that further supports our observations.

With this new dataset we have substantially reorganized the manuscript; first we described the bulk transcriptome cohort, which is followed by the second cohort analyzed with the higher resolution single cell RNAseq. By combining two independent cohorts with different levels of resolution, the revised manuscript provides an enhanced longitudinal systems-level analysis of the immune system in vaccinated young infants.

The new data were obtained from an independent infant cohort that received their routine 2-month-old vaccinations. Data were collected in a manner similar to the first cohort, both before vaccination (day 0) and approximately one week (day 7) and one month (day 30) later. We have extensively revised the manuscript, and the findings of the new analysis are summarized in the following additional figures:

- **Supplementary Figure 1A: serologic studies;**
- **Figure 1A: Heatmap of the bulk transcriptome studies, and**

- **Figure 1B: Deconvolution of the whole blood transcriptome using a modular analytical tool to define transcriptional changes in genes related to major immune pathways and cell populations.**

In summary, we believe that this additional dataset derived from an independent infant cohort applying a similar study design, although using different and complementary analytical tools, validates and extend our findings: 1) It illustrates the variability of the antibody responses of these young infants; 2) it provides evidence of a robust ISG response at day 7 (resolved by day 30), which as demonstrated by scRNAseq involves most immune cell populations; 3) the bulk analysis also identified early overexpression of inflammation-related genes (increased expression of CCR1 and TNFSF13B within M5.1) that was confirmed by single cell analysis in CD16 monocytes; and 4) it showed increased expression of modules corresponding to B cell and plasma cell genes at 4 weeks post-vaccination, when the scRNA-seq analysis showed an increase frequency of B_{naive} and B_{memory} cells suggesting an expansion and/or more persisting activation of the B cell compartment.

REVIEWERS' COMMENTS

Reviewer #1 (expert in transcriptomics):

My comments have been satisfactorily addressed. The inclusion of the bulk transcriptomes cohort has substantially enhanced the overall quality of the work, and I commend the effort to incorporate it. Several key findings remain consistent and have been appropriately emphasized, such as those pertaining to interferon/ISGs on Day 7 and B cells on Day 30. One minor question I still have pertains to the reduction of the T cell module in the bulk cohort on Day 7 and how you would reconcile this with the results from the single-cell cohort. However, I believe that a sentence or two in the discussion section to address this will suffice.

Overall, I recommend the publication of this manuscript.

Reviewer #1, commenting on the responses to Reviewer #3, who was no longer available for review:

Considering the challenges associated with obtaining data from cohorts involving the longitudinal collection of blood samples from infants, it is valuable to publish the data, provided that the authors exercise caution in drawing conclusions. The revised manuscript reflects the authors' improved carefulness in their claims and a transparent acknowledgment of the study's limitations. Furthermore, the inclusion of results from the additional 'bulk' cohort has strengthened their general claims regarding the overall immune cell landscape, partially addressing the concerns raised by Reviewer #3.

However, it is still important for the authors to acknowledge another limitation brought up by Reviewer #3: "the lack of potentially any vaccine specific cells in the results." They could highlight the need to explore this aspect in future research, either by themselves or others.

Regarding the deeper sequencing of the original samples, a complete redo of the project may not be a practical option for the authors. Given the incremental contributions of their work, which is based on challenging cohorts to acquire, the findings and conclusions thus far merit publication. The authors could also underscore the necessity of future experiments and the significance of their current work in justifying these future investigations. For example, given the observed variability in responses and some evidence of associated underlying immune components, their research could serve as a compelling basis for exploring the potential need for more personalized vaccination strategies in infants.

Reviewer #2 (expert in infectious disease paediatrics):

The authors have addressed my concerns on the revised manuscript. The manuscript is significantly improved, especially with the data from the additional second cohort.

Reviewer #3 (expert in 'Systems' Immunology and inborn errors or immunity):

No longer available for review.

REVIEWERS' COMMENTS

Reviewer #1 (expert in transcriptomics):

My comments have been satisfactorily addressed. The inclusion of the bulk transcriptomes cohort has substantially enhanced the overall quality of the work, and I commend the effort to incorporate it. Several key findings remain consistent and have been appropriately emphasized, such as those pertaining to interferon/ISGs on Day 7 and B cells on Day 30. One minor question I still have pertains to the reduction of the T cell module in the bulk cohort on Day 7 and how you would reconcile this with the results from the single-cell cohort. However, I believe that a sentence or two in the discussion section to address this will suffice.

Overall, I recommend the publication of this manuscript.

Response to Reviewer #1

We appreciate Reviewer #1 comments. We added the following sentence to the discussion:

“Also, on day 7 the bulk transcriptome analysis showed mild underexpression of T cell genes, likely reflecting the marked difference with the prominent overexpression of interferon and inflammation genes present at that time. Similar observations have been previously reported in transcriptome analysis performed in children with acute viral infections (Mejias, 2013 #7). This potential limitation of the bulk transcriptome can now be addressed with the higher resolution provided by the single-cell analysis”.

Reviewer #1, commenting on the responses to Reviewer #3, who was no longer available for review:

Considering the challenges associated with obtaining data from cohorts involving the longitudinal collection of blood samples from infants, it is valuable to publish the data, provided that the authors exercise caution in drawing conclusions. The revised manuscript reflects the authors' improved carefulness in their claims and a transparent acknowledgment of the study's limitations. Furthermore, the inclusion of results from the additional 'bulk' cohort has strengthened their general claims regarding the overall immune cell landscape, partially addressing the concerns raised by Reviewer #3.

However, it is still important for the authors to acknowledge another limitation brought up by Reviewer #3: “the lack of potentially any vaccine specific cells in the results.” They could highlight the need to explore this aspect in future research, either by themselves or others.

Regarding the deeper sequencing of the original samples, a complete redo of the project may not be a practical option for the authors. Given the incremental contributions of their work, which is based on challenging cohorts to acquire, the findings and conclusions thus far merit publication. The authors could also underscore the necessity of future experiments and the

significance of their current work in justifying these future investigations. For example, given the observed variability in responses and some evidence of associated underlying immune components, their research could serve as a compelling basis for exploring the potential need for more personalized vaccination strategies in infants.

Response to Reviewer #1 in relation to the comments of Reviewer #3

We appreciate these comments and the following sentences have been added to the discussion.

... and did not allow us to examine vaccine-specific immune cells...

The significant variation observed both in antibody and immune cell responses highlights the need for future studies focused in young infants. Further, it could argue towards development of more personalized vaccination strategies for this population.

Reviewer #2 (expert in infectious disease paediatrics):

The authors have addressed my concerns on the revised manuscript. The manuscript is significantly improved, especially with the data from the additional second cohort.

We appreciate Reviewer #2 comments.

Reviewer #3 (expert in 'Systems' Immunology and inborn errors or immunity):

No longer available for review.